# Inverse Approximation Theory for Nonlinear Recurrent Neural Networks

**Shida Wang**
Department of Mathematics
National University of Singapore
e0622338@u.nus.edu

**Zhong Li**
Microsoft Research Asia
lzhong@microsoft.com

**Qianxiao Li**[*]
Department of Mathematics
Institute for Functional Intelligent Materials
National University of Singapore
qianxiao@nus.edu.sg

## Abstract

We prove an inverse approximation theorem for the approximation of nonlinear sequence-to-sequence relationships using recurrent neural networks (RNNs). This is a so-called Bernstein-type result in approximation theory, which deduces properties of a target function under the assumption that it can be effectively approximated by a hypothesis space. In particular, we show that nonlinear sequence relationships that can be stably approximated by nonlinear RNNs must have an exponential decaying memory structure - a notion that can be made precise. This extends the previously identified curse of memory in linear RNNs into the general nonlinear setting, and quantifies the essential limitations of the RNN architecture for learning sequential relationships with long-term memory. Based on the analysis, we propose a principled reparameterization method to overcome the limitations. Our theoretical results are confirmed by numerical experiments.

## 1 Introduction

Recurrent neural networks (RNNs) (Rumelhart et al., 1986) are one of the most basic machine learning models to learn the relationship between sequential or temporal data. They have wide applications from time series prediction (Connor et al., 1994), text generation (Sutskever et al., 2011), speech recognition (Graves & Jaitly, 2014) to sentiment classification (Tang et al., 2015). However, when there are long-term dependencies in the data, empirical results (Bengio et al., 1994) show that RNN may encounter difficulties in learning. In this paper, we investigate this problem from the view of approximation theory.

From approximation perspective, there are various types of theorems characterizing the connections between target relationships and model architectures for learning them. Universal approximation (Achieser, 2013, p. 32) and Jackson-type theorem (Achieser, 2013, p. 187) provide basic guarantees of approximation and error estimates of sufficiently regular target functions by a particular hypothesis space. A number of such results are available for sequence modeling, including the RNN (Li et al., 2020; 2022). On the other hand, a relatively under-investigated domain in the machine learning literature are Bernstein-type theorems (Bernstein, 1914; Li et al., 2022), which are also known as inverse approximation theorems. These results aims to characterize the regularity of target relationships, assuming that they can be approximated efficiently with a hypothesis space. These regularity notions intimately depends on, thus gives important insights into, the structure of the hypothesis space under study.

---

[*]Corresponding author

This paper establishes an inverse approximation result for the approximation of nonlinear functionals via RNNs. Previous works (Li et al., 2020; 2022) indicate that linear functionals that can be universally approximated by linear RNNs must have exponential decaying memory. This phenomena was coined the *curse of memory* for linear RNNs. A natural question is whether the nonlinear recurrent activation used in practical RNNs changes the situation. This is important since a bigger hypothesis space may lift restrictions on the target functions. Moreover, it is known that nonlinear activation is crucial for feed-forward networks to achieve universality (Cybenko, 1989). Thus, it is worthwhile to investigate whether the linear Bernstein result generalizes to the case of approximating nonlinear sequence relationships with nonlinear RNNs. In this paper, we prove that nonlinear RNNs still suffer from a curse of memory in approximation - nonlinear functionals that can be stably approximated by RNNs with nonlinear activations must have an exponential decaying memory function. The notions of stable approximation and memory function can be concretely defined. Our results make precise the empirical observation that the RNN architecture has inherent limitations when modeling long-time dependencies.

In summary, our main contributions are:

1. We extends the concept of memory function in the linear settings (Li et al., 2020; 2022) to the nonlinear setting. This memory function can be numerically quantified in sequence modeling applications.

2. We introduce a notion of stable approximation, which ensures that an approximant has the possibility to be found by a gradient-based optimization algorithm.

3. We prove, to the best of our knowledge, the first Bernstein-type approximation theorem for nonlinear functional sequences through nonlinear RNNs. Our results characterize the essential limit of nonlinear RNNs in learning long-term relationship. Our analysis also suggests that appropriate parameterization can alleviate the 'curse of memory' phenomenon in learning targets with long memory. The theoretical result is corroborated with numerical experiments.

**Notation.** For a sequence of $d$-dimensional vectors indexed by $\mathbb{R}$, $\mathbf{x} = \{x_t \in \mathbb{R}^d : t \in \mathbb{R}\}$, we denote the supremum norm by $\|\mathbf{x}\|_\infty := \sup_{t \in \mathbb{R}} |x_t|_\infty$. Here $|x|_\infty := \max_i |x_i|, |x|_2 := \sqrt{\sum_i x_i^2}, |x|_1 := \sum_i |x_i|$ are the usual max ($L_\infty$) norm, $L_2$ norm and $L_1$ norm. The bold face represents sequence while the normal letters are scalars, vectors or functions. Throughout this paper we use $\|\cdot\|$ to denote norms over sequences of vectors, or function(al)s, while $|\cdot|$ (with subscripts) represents the norm of number, vector or weights tuple. The hat notation in this paper refers to the hypothesis space (functional) while the original symbol is referring to the target space (functional).

## 2 PROBLEM FORMULATION AND PRIOR RESULTS ON LINEAR RNNS

In this section, we introduce the problem formulation of sequence modeling as a functional sequence approximation problem (Jiang et al., 2023). We pay particular attention to distinguish two types of results: forward (Jackson-type) and inverse (Bernstein-type) approximation theorems. For approximation theory in machine learning, most existing results focus on forward theorems. However, inverse approximation theorems are of significant importance in revealing the fundamental limitations of an approximation approach. This paper focuses on establishing such results in the general, non-linear setting. We conclude this section with a review of known Bernstein-type estimates, which is currently restricted to the linear case. In so doing, we highlight the definition of memory in the linear case, which motivates our general definition of memory for nonlinear functional sequences in Section 3.1. The relationship between memory and approximation is central to our results.

### 2.1 THE APPROXIMATION PROBLEM FOR SEQUENCE MODELING

The goal of sequential modeling is to learn a relationship between an input sequence $\mathbf{x} = \{x_t\}$ and a corresponding output sequence $\mathbf{y} = \{y_t\}$. For ease of analysis, we adopt the continuous-time setting in (Li et al., 2020), where $t \in \mathbb{R}$. This is also a natural setting for irregularly sampled time series (Lechner & Hasani, 2020). The input sequence space is $\mathcal{X} = C_0(\mathbb{R}, \mathbb{R}^d)$, the space of continuous functions vanishing at infinity. We assume the input and output sequences are related by a sequence of functionals $\mathbf{H} = \{H_t : \mathcal{X} \mapsto \mathbb{R}; t \in \mathbb{R}\}$ via $y_t = H_t(\mathbf{x}), t \in \mathbb{R}$. The sequential

approximation problem can be formulated as the approximation of the target functional sequence $\mathbf{H}$ by a functional sequence $\widehat{\mathbf{H}}$ from a model hypothesis space such as RNNs.

**Forward and inverse approximation theorems.** Given a hypothesis space $\widehat{\mathcal{H}}^{(m)}$ of complexity $m \geq 1$ (e.g. width-$m$ RNNs), forward approximation theorems, also called Jackson-type theorems, bound the optimal approximation error $\inf_{\widehat{\mathbf{H}} \in \widehat{\mathcal{H}}^{(m)}} \|\mathbf{H} - \widehat{\mathbf{H}}\| \leq C(\mathbf{H}, m)$.

Inverse approximation (Bernstein-type) results are "converse" statements to Jackson-type results. From the starting assumption that an efficient approximation exists for a specific target $\mathbf{H}$, Bernstein-type results deduce the approximation spaces that $\mathbf{H}$ ought to belong to, i.e. it identifies a complexity or regularity measure $C(\cdot)$ and show that $C(\mathbf{H})$ is necessarily finite. Take the trigonometric polynomial approximation as an example, Bernstein (Achieser, 2013, p. 187) proved that if $\inf_{\widehat{H} \in \widehat{\mathcal{H}}^{(m)}} \|H - \widehat{H}\| \leq \frac{c}{m^{\alpha+\delta}}$, for all $m \geq 1$ and some $\delta > 0$, $c > 0$, then $C(H) = |H^{(\alpha)}| < \infty$, i.e. $H$ must be $\alpha$-times differentiable with $\delta$-Hölder continuous derivatives.

Bernstein-type inverse approximation results are important in characterizing the approximation capabilities for hypothesis spaces. For trigonometric polynomial example, it says that *only* smooth functions can be efficiently approximated, thereby placing a concrete limitation on the approximation capabilities of these models. Our goal in this paper is to deduce analogues of this result, but for the approximation of general nonlinear functional sequences by RNNs. Unlike the classical case where the notion of regularity enters in the form of smoothness, here we shall investigate the concept of memory as a quantifier of regularity - a notion that we will make precise subsequently.

## 2.2 THE RNN ARCHITECTURE AND PRIOR RESULTS

The continuous-time RNN architecture parameterizes functional sequences by the introduction of a hidden dynamical system

$$\begin{array}{ll} \frac{dh_t}{dt} & = \sigma(W h_t + U x_t + b) \quad h_{-\infty} = 0, \\ \hat{y}_t & = c^\top h_t, \quad t \in \mathbb{R}. \end{array} \tag{1}$$

Here, $\hat{y}_t \in \mathbb{R}$ is the predicted output sequence value, and $h_t \in \mathbb{R}^m$ denotes the hidden state[1]. The well-definedness of the continuous-time RNN will be discussed in Appendix A.1. The hyperparameter $m$ is also known as the hidden dimension, or width, of recurrent neural networks. For different hidden dimensions $m$, the RNN is parameterized by trainable weights $(W, U, b, c)$, where $W \in \mathbb{R}^{m \times m}$ is the recurrent kernel, $U \in \mathbb{R}^{m \times d}$ is the input kernel, $b \in \mathbb{R}^m$ is the bias and $c \in \mathbb{R}^m$ is the readout. The complexity of the RNN hypothesis space is characterized by the hidden dimension $m$. The nonlinearity arises from the activation function $\sigma(\cdot)$, which is a scalar function performed element-wise, such as *tanh*, *hardtanh*, *sigmoid* or *ReLU*. The hypothesis space of RNNs is thus the following functional sequence space

$$\widehat{\mathcal{H}}_{\text{RNN}}^{(m)} = \{\mathbf{x} \mapsto \hat{\mathbf{y}} \text{ via Equation (1)} : W \in \mathbb{R}^{m \times m}, U \in \mathbb{R}^{m \times d}, b \in \mathbb{R}^m, c \in \mathbb{R}^m\}. \tag{2}$$

Before presenting our main results, we review known Jackson and Bernstein-type results established for linear RNNs, corresponding to setting $\sigma(z) = z$ and $b = 0$ in equation 1. We shall pay attention to the definition of memory for a target functional sequence, and how it relates to approximation properties under the RNN hypothesis space.

We begin with some definitions on (sequences of) functionals as introduced in (Li et al., 2020).

**Definition 2.1.** Let $\mathbf{H} = \{H_t : \mathcal{X} \mapsto \mathbb{R}; t \in \mathbb{R}\}$ be a sequence of functionals. Without loss of generality, we assume $H_t(\mathbf{0}) = 0$ (Otherwise we can consider the adjusted relationship $H_t^{\text{adjusted}}(\mathbf{x}) := H_t(\mathbf{x}) - H_t(\mathbf{0})$).

1. (**Linear**) $H_t$ is linear if for any $\lambda, \lambda' \in \mathbb{R}$ and $\mathbf{x}, \mathbf{x}' \in \mathcal{X}$, $H_t(\lambda \mathbf{x} + \lambda' \mathbf{x}') = \lambda H_t(\mathbf{x}) + \lambda' H_t(\mathbf{x}')$.

---

[1] The boundary condition $h_{-\infty} = 0$ is consistent with practical implementations such as TensorFlow and PyTorch, where the initial value of hidden state is set to be zero by default.

2. (**Continuous**) $H_t$ is continuous if for any $\mathbf{x},' \mathbf{x} \in \mathcal{X}$, $\lim_{\mathbf{x}' \to \mathbf{x}} |H_t(\mathbf{x}') - H_t(\mathbf{x})| = 0$.

3. (**Bounded**) $H_t$ is bounded if $\sup_{\{\mathbf{x} \in \mathcal{X}, \mathbf{x} \neq 0\}} \frac{|H_t(\mathbf{x})|}{\|\mathbf{x}\|_\infty} < \infty$.

4. (**Time-homogeneous**) $\mathbf{H} = \{H_t : t \in \mathbb{R}\}$ is time-homogeneous (or shift-equivariant) if the input-output relationship commutes with time shift: let $[S_\tau(\mathbf{x})]_t = x_{t-\tau}$ be a shift operator, then $\mathbf{H}(S_\tau \mathbf{x}) = S_\tau \mathbf{H}(\mathbf{x})$

5. (**Causal**) $H_t$ is causal if it does not depend on future values of the input. That is, if $\mathbf{x}, \mathbf{x}'$ satisfy $x_t = x'_t$ for any $t \leq t_0$, then $H_t(\mathbf{x}) = H_t(\mathbf{x}')$ for any $t \leq t_0$.

6. (**Regular**) $H_t$ is regular if for any sequence $\{\mathbf{x}^{(n)} : n \in \mathbb{N}\}$ such that $x_s^{(n)} \to 0$ for almost every $s \in \mathbb{R}$, then $\lim_{n \to \infty} H_t(\mathbf{x}^{(n)}) = 0$.

The works in Li et al. (2020; 2022) study the approximation of functional sequences satisfying Definition 2.1 by linear RNNs. A key idea is showing that any such functional sequence $\mathbf{H}$ admits a Riesz representation (see Appendix A.2 and Appendix A.3)

$$H_t(\mathbf{x}) = \int_0^\infty \rho(s)^\top x_{t-s} ds, \qquad t \in \mathbb{R}. \tag{3}$$

In this sense, $\rho$ completely determines $\mathbf{H}$, and its approximation using linear RNNs can be reduced to the study of the approximation of $\rho \in L^1([0, \infty), \mathbb{R}^d)$ by exponential sums of the form $(c^\top e^{Ws} U)^\top$. An important observation here is that $\rho$ captures the memory pattern of the target linear functional sequence: if $\rho$ decays rapidly, then the target has short memory, and vice versa.

By assuming that a target functional sequence $\mathbf{H}$ can be approximated uniformly by stable RNNs, then the memory of the target functional sequence must satisfy $e^{\beta_0 t} |\rho(t)|_1 = o(1)$ as $t \to \infty$ for some $\beta_0 > 0$. This was coined the "curse of memory" (Li et al., 2020; 2022) and reveals fundamental limitations of the RNN architecture to capture long-term memory structures.

The focus of this paper is to investigate whether the addition of nonlinear activation changes this result. In other words, would the curse of memory hold for nonlinear RNNs in the approximation of suitably general nonlinear functionals? This is a meaningful question, since Bernstein-type results essentially constrain approximation spaces, and so a larger hypothesis space may relax such constraints. We expand on this in Appendix A.4. A significant challenge in the nonlinear setting is the lack of a Riesz representation result, and thus one needs to carefully define a notion of memory that is consistent with $\rho$ in the linear case, but can still be used in the nonlinear setting to prove inverse approximation theorems. Moreover, we will need to introduce a general notion of approximation stability, which together with the generalized memory definition allows us to derive a Bernstein-type result that holds beyond the linear case.

## 3 MAIN RESULTS

In this section, we establish a Bernstein-type approximation result for nonlinear functional sequences using nonlinear RNNs. We first give a definition of memory function for nonlinear functionals. It is compatible with the memory definition in the linear functionals and it can be queried and verified in applications. Next, we propose the framework of stable approximation. It is a mild requirement from the perspective of approximation, but a desirable one from the view of optimization. Moreover, we show that any linear functional with an exponential decaying memory can be stably approximated. Based on the memory function definition and stable approximation framework, we prove a Bernstein-type theorem. The theorem shows that any nonlinear functionals that can be stably approximated by general nonlinear RNNs must have an exponentially decaying memory, which confirms that the curse-of-memory phenomenon is not limited to the linear case. Numerical verifications are included to demonstrate the result.

### 3.1 MEMORY FUNCTION FOR NONLINEAR FUNCTIONALS

Recall that the memory for a linear functional sequence is defined by its Riesz representation in Equation (3). While there are no known general analogues of Riesz representation for nonlinear functionals, we may consider other means to extract an effective memory function from $\mathbf{H}$.

Let $x \in \mathbb{R}^d$ and consider the following Heaviside input sequence $\mathbf{u}_t^x = x \cdot \mathbf{1}_{[0,\infty)}(t) = \begin{cases} x & t \geq 0, \\ 0 & t < 0. \end{cases}$

In the linear case, notice that according to Equation (3)

$$\sup_{x \neq 0} \frac{\left| \frac{d}{dt} H_t(\mathbf{u}^x) \right|}{\|\mathbf{u}^x\|_\infty} = \sup_{x \neq 0} \frac{|x^\top \rho(t)|}{|x|_\infty} = |\rho(t)|_1. \tag{4}$$

Hence, conditions on $|\rho(t)|_1$ may be replaced by conditions on the left hand side, which is well-defined also for nonlinear functionals. This motivates the following definition of memory function for nonlinear functional sequences.

**Definition 3.1** (Memory function of nonlinear functional sequences). For continuous, causal, regular and time-homogeneous functional sequences $\mathbf{H} = \{H_t(\mathbf{x}) : t \in \mathbb{R}\}$ on $\mathcal{X}$, define the following function as the *memory function* of $\mathbf{H}$ over bounded Heaviside input $\mathbf{u}^x$:

$$\mathcal{M}(\mathbf{H})(t) := \sup_{x \neq 0} \frac{1}{|x|_\infty} \left| \frac{d}{dt} H_t(\mathbf{u}^x) \right|. \tag{5}$$

In particular, in this paper we consider nonlinear functionals whose memory function is finite for all $t$. Unlike traditional methods that evaluate memory through heuristic tasks, our approach offers a precise, task-independent characterization of model memories. If the oracle of the target functional is available, the memory function can be evaluated and the result is named queried memory. In Appendix F and Appendix G, we discuss the memory function evaluated over different test inputs and show the numerical equivalence in Appendix H. Without target functional oracle, we may approximate the target with the learned model and still evaluate the memory function. If the queried memory are decaying for all Heaviside inputs, then we say the corresponding nonlinear functional sequence has a decaying memory. We demonstrate in Appendix B that the memory querying shows the memory pattern of LSTM and bidirectional LSTM sequence-to-sequence models in sentiment analysis on IMDB movie reviews.

**Definition 3.2** (Decaying memory). For continuous, causal, regular and time-homogeneous functional sequences $\mathbf{H} = \{H_t(\mathbf{x}) : t \in \mathbb{R}\}$ on $\mathcal{X}$, we say it has a *decaying memory* if:

$$\lim_{t \to \infty} \mathcal{M}(\mathbf{H})(t) = 0. \tag{6}$$

We say that $\mathbf{H}$ has an *exponential decaying memory* if for some $\beta > 0$,

$$\lim_{t \to \infty} e^{\beta t} \mathcal{M}(\mathbf{H})(t) = 0. \tag{7}$$

Furthermore, the family $\{\mathbf{H}_m\}$ has a *uniformly decaying memory* if

$$\lim_{t \to \infty} \sup_m \mathcal{M}(\mathbf{H}_m)(t) = 0. \tag{8}$$

*Remark* 3.3. The requirement of decaying memory on time-homogeneous functionals is mild since it is satisfied if $\frac{dH_t}{dt}$ is continuous at Heaviside input, under the topology of point-wise convergence (see Appendix A.5). We show that $\frac{dH_t}{dt}$ are point-wise continuous over Heaviside inputs for all RNNs, thus RNNs have decaying memory (see Appendix A.6). Another related notion of fading memory is discussed in the Appendix A.7.

## 3.2 Stable approximation

We now introduce the stable approximation framework. Let us write the hypothesis space $\widehat{\mathcal{H}}^{(m)}$ as a parametric space $\widehat{\mathcal{H}}^{(m)} = \{\widehat{\mathbf{H}}(\cdot; \theta_m) : \theta_m \in \Theta_m\}$ where for each $m$, $\Theta_m$ is a subset of a Euclidean space with dimension depending on $m$, representing the parameter space defining the hypothesis and $\widehat{\mathbf{H}}$ is a parametric model. For example, in the case of RNNs, the parameter $\theta_m$ is $(W_m, U_m, b_m, c_m) \in \Theta_m := \{\mathbb{R}^{m \times m} \times \mathbb{R}^{m \times d} \times \mathbb{R}^m \times \mathbb{R}^m\}$ and $m$ is the hidden dimension of the RNN.

Let us consider a collection of functional sequences $\{\widehat{\mathbf{H}}_m = \widehat{\mathbf{H}}(\cdot; \theta_m) : m \geq 1\}$ that approximates a target functional sequence $\mathbf{H}$. Stable approximation requires that, if one were to perturb each parameter $\theta_m$ by a small amount, the resulting approximant sequence should still have a continuous perturbation error. For the gradient-based optimization, this condition is necessary for one to find such an approximant sequence, as small perturbations of parameters should keep perturbation error continuous for gradients to be computed. We now define this notion of stability precisely.

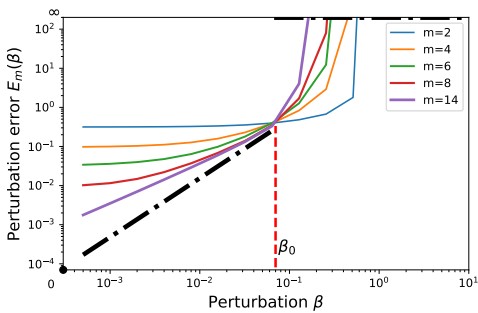 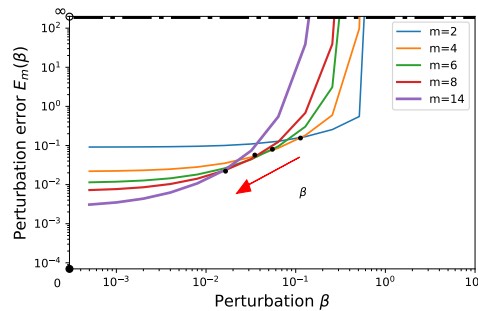

(a) Target with exponential decay memory          (b) Target with polynomial decay memory

Figure 1: Perturbation errors for linear functionals with different decaying memory. The anticipated limiting curve $E(\beta)$ is marked with a black dashed line. (a) For linear functional sequences with exponential decaying memory, there exists a perturbation radius $\beta_0$ such that the perturbation error $E(\beta)$ for $0 \leq \beta < \beta_0$ is continuous. (b) Approximation of linear functional sequences with polynomial decaying memory. As hidden dimension $m$ increases, the perturbation radius where the error remains small decreases, suggesting that there may not exist a $\beta_0$ achieving the stable approximation condition. The intersections of lines are shifting left as the hidden dimension $m$ increases. The anticipated limiting curve $E(\beta)$ is not continous for the polynomial decaying memory target.

**Definition 3.4.** For target $\mathbf{H}$ and parameterized model $\widehat{\mathbf{H}}(\cdot, \theta_m)$, we define the perturbation error for hidden dimension $m$ to be:

$$E_m(\beta) := \sup_{\tilde{\theta}_m \in \{\theta : |\theta - \theta_m|_2 \leq \beta\}} \|\mathbf{H} - \widehat{\mathbf{H}}(\cdot; \tilde{\theta}_m)\| \tag{9}$$

Moreover, $E(\beta) := \limsup_{m \to \infty} E_m(\beta)$ is the (asymptotic) perturbation error. Here $|\theta|_2 := \max(|W|_2, |U|_2, |b|_2, |c|_2)$.

**Definition 3.5** (Stable approximation via parameterized models)**.** Let $\beta_0 > 0$. We say a target functional sequence $\mathbf{H}$ admits a $\beta_0$-stable approximation under $\{\widehat{\mathcal{H}}^{(m)}\}$, if there exists a sequence of parameterized approximants $\widehat{\mathbf{H}}_m = \widehat{\mathbf{H}}(\cdot, \theta_m), \theta_m \in \Theta_m$ such that

$$\lim_{m \to \infty} \|\mathbf{H} - \widehat{\mathbf{H}}_m\| \to 0, \tag{10}$$

and for all $0 \leq \beta \leq \beta_0$, the perturbation error satisfies that $E(\beta)$ is continuous in $\beta$ for $0 \leq \beta \leq \beta_0$.

*Remark* 3.6. It can be seen that approximation only requires $E(0) = 0$. Therefore the stable approximation condition generalizes the approximation by requiring the continuity of $E$ around $\beta = 0$. If an approximation is unstable ($E(0) = 0, \lim_{\beta \to 0} E(\beta) > 0$), it is difficult to be found by gradient-based optimizations. Since our notion of stability depends on the size of weight perturbations, one may wonder whether rescaling the norm of weights separately for each $m$ achieves stability. There are two issues with this approach. First, the rescaled version is no-longer the usual RNN hypothesis space. Second, to achieve stability the rescaling rule may depend on information of the target functional sequence, which we have no access to in practice. We discuss this in detail in Appendix C.

Next, we demonstrate that the stable approximation condition is not too stringent in the sense that, for linear functional sequence with exponential decaying memory (Equation (7)) admits a stable approximation. We show the numerical verification of this result in Figure 1. The approximation of linear functional with exponential decay can be seen in the left panel at $\beta = 0$ since increasing the hidden dimension $m$ will make the estimated error decrease to 0 over $\beta \in [0, \beta_0]$. Stable approximation can be verified that for positive perturbation $\beta$, adding the hidden dimension does not increase the perturbation error $E(\beta)$. In contrast, for linear functional with polynomial decaying memory, the perturbation error $E(\beta)$ is not continuous at $\beta = 0$.

### 3.3 BERNSTEIN-TYPE APPROXIMATION RESULT FOR NONLINEAR RNNS

We now present the main result of this paper, which is a Bernstein-type approximation result for nonlinear functional sequences using nonlinear RNNs. The key question is whether the addition

of nonlinearity alleviates the curse of memory limitation and allows an efficient approximation of functionals with slow memory decay. In the following, we show that the answer is negative, and a similar Bernstein-type approximation result holds for nonlinear functionals and RNNs with a class of recurrent activations, including the most often used hardtanh/tanh activations.

**Definition 3.7.** We consider the Sobolev-type norm:

$$\left\|\mathbf{H} - \widehat{\mathbf{H}}\right\|_{W^1} = \sup_t \left( \|H_t - \widehat{H}_t\|_\infty + \left\|\frac{dH_t}{dt} - \frac{d\widehat{H}_t}{dt}\right\|_\infty \right). \tag{11}$$

The nonlinear functional norm is given by $\|H_t\|_\infty := \sup_{\mathbf{x}\neq 0} \frac{|H_t(\mathbf{x})|}{\|\mathbf{x}\|_\infty} + |H_t(\mathbf{0})| = \sup_{\mathbf{x}\neq 0} \frac{|H_t(\mathbf{x})|}{\|\mathbf{x}\|_\infty}$.

**Definition 3.8.** We consider the following family of **bounded monotone Lipschitz continuous** activations which are locally-linear/locally-tanh around 0: For some $Z_0 > 0$,

$$\mathcal{A}_0 := \{\sigma(\cdot)|\sigma(z) = c_\sigma z, c_\sigma > 0, |z| < Z_0\}, \tag{12}$$

$$\mathcal{A}_1 := \{\sigma(\cdot)|\sigma(0) = 0; \sigma \text{ differentiable}, \sigma'(z) = a - b\sigma(z)^2, a, b \geq 0, |z| < Z_0\}. \tag{13}$$

Notice $\mathcal{A}_0 \cup \mathcal{A}_1$ includes the commonly used activations such as hardtanh and tanh. In particular, tanh corresponds to the case $a = b = 1$ for $\mathcal{A}_1$ with $Z_0 = \infty$.

**Theorem 3.9.** *Assume $\mathbf{H}$ is a sequence of bounded continuous, causal, regular and time-homogeneous functionals on $\mathcal{X}$ with decaying memory. Let the activation be in $\mathcal{A}_0 \cup \mathcal{A}_1$. Suppose $\mathbf{H}$ is $\beta_0$-stably approximated by a sequence of RNNs $\{\widehat{\mathbf{H}}(\cdot, \theta_m)\}_{m=1}^\infty$ in the norm defined in Equation (11). If the perturbed models' memory functions are uniformly decaying (as defined in Definition 3.2) and the weight norms are uniformly bounded in $m$:*

$$\sup_m |\theta_m|_2 < \infty. \tag{14}$$

*Then the memory function $\mathcal{M}(\mathbf{H})(t)$ of the target decays exponentially:*

$$\lim_{t\to\infty} e^{\beta t}\mathcal{M}(\mathbf{H})(t) = 0, \quad \beta < \beta_0. \tag{15}$$

The proofs are included in the Appendix A.8. Given that approximations are required to be stable, the decaying memory property ensures that the derivative of the hidden states for the perturbed model approaches 0 as time $t \to \infty$. Using the Hartman-Grobman theorem, we can obtain bounds on the eigenvalues of the matrices $W_m$. In Appendix J, we demonstrate that our methods can be generalized to analyze the dynamics of GRU and LSTM. The framework is similar while the final hidden dynamics of GRU and LSTM require more techniques to analyze.

**Interpretation of Theorem 3.9.** Our main result (Theorem 3.9) extends the previous linear result from Li et al. (2022). Instead of smoothness (measured by the Sobolev norm) as a regularity measure, the RNN Bernstein-type result identifies exponential decaying memory ($e^{\beta t}\mathcal{M}(\mathbf{H})(t) \to 0$) as the right regularity measure. If we can approximate some target functionals stably using nonlinear RNN, then that target must have exponential decaying memory. Previously this was only known for linear case, but for nonlinear case, even addition of nonlinearity substantially increases model complexity, it does not fix the essential memory limitation of RNNs.

From the numerical perspective, the theorem implies the following two statements, and we provide numerical verification for each of them. First, if the memory function of a target functional sequence decays slower than exponential (e.g. $\mathcal{M}(\mathbf{H})(t) = \frac{C}{(t+1)^{1.5}}$), the optimization is difficult and the approximation in Figure 2 is achieved at 1000 epochs while typically exponential decaying memory achieves the approximation at 10 epochs. When the approximation is achieved, it can be seen in Figure 2 that, for larger perturbation scale $\beta$, there is no perturbation stability. Second, if a target functional sequence can be well-approximated and the approximation's stability radius $\beta_0$ can be shown to be positive, then the target functional sequence should have exponential decaying memory. See Figure 3 for the approximation filtered with perturbation stability requirement. (See Figure 5 in Appendix B for the validation of memory over general sentiment classification task.)

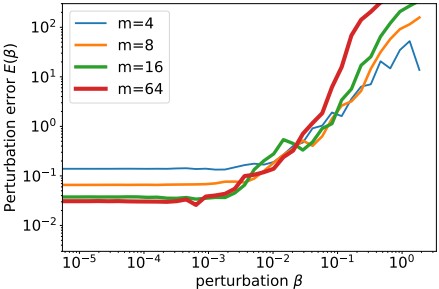

Figure 2: Target with polynomial decaying memory + approximation (achieved at 1000 epochs) $\rightarrow$ no stability. Similar to the linear functional case, when approximating nonlinear functionals with polynomial decaying memory by tanh RNN, the intersections of curves are shifting left as the hidden dimension $m$ increases.

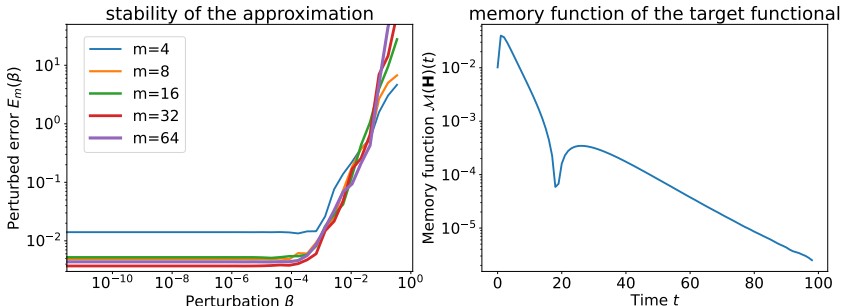

Figure 3: Stable approximation via RNNs implies exponential decaying memory. We construct several randomly-initialized RNN models as teacher models with large hidden dimension ($m = 256$). When approximating the teacher model with a series of student RNN models, we can numerically verify the approximation's stability (left panel). We can apply a filtering: we only select those teacher models which both can be approximated, and the approximations are stable (with perturbation error $E_m(\beta)$ having a positive stability radius). We found that the only teachers that remain are those with exponential decaying memory functions. An example is shown in the right panel.

### 3.4 SUITABLE PARAMETRIZATION ENABLES STABLE APPROXIMATION

The key insight of Theorem 3.9 can be summarized as follow: in order to approximate targets with non-exponential decaying memory, the recurrent weights of RNNs must have eigenvalues' real part approaching 0 on the negative side. However, if the largest eigenvalue real parts are approaching zero, then its stability under perturbation will decrease. This is why the approximation and stability cannot be achieved at the same time if the target's memory does not decay exponentially. The stability problem can be resolved via reparameterization as the stability radius is not decreasing even when the eigenvalues are approaching 0. If we reparameterize the recurrent weights so that it approach zero and remain stable (i.e., eigenvalue real part being negative) under perturbations, then this architecture will maintain stability while having the possibility of approximation. We can accomplish this by substituting the recurrent weight with a continuous matrix function, which we will refer to as "stable reparameterization"

$$g : \mathbb{R}^{m \times m} \to \mathbb{R}^{m \times m, -}, \quad g(M) = W. \tag{16}$$

This reparameterized RNN is stable as the eigenvalues' real part are always negative. We show there are several methods to achieve this reparameterization: The exponential function $g(M) = -e^M$ and the softplus function $g(M) = -\log(1 + e^M)$ maps the eigenvalues of $M$ to negative range (see Figure 4 and Figure 8 for the stable approximation of linear functional with polynomial decay memory). LRU (Orvieto et al., 2023) proposed to parameterizes the real part of eigenvalues by $\exp(-\exp(\lambda))$, which corresponds to the discrete case for $g(M) = -e^M$.

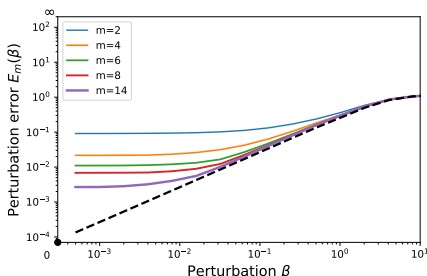 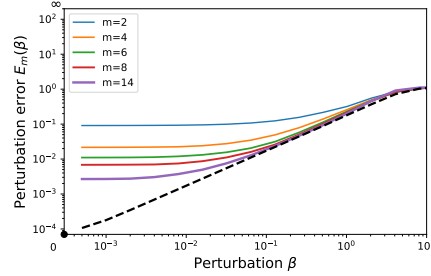

(a) Exp reparameterization + Linear RNN      (b) Softplus reparameterization + Linear RNN

Figure 4: Stable approximation of linear functionals with polynomial decay memory by linear RNN with exp and softplus reparameterization. The limiting dashed curve $E(\beta)$ shall be continuous.

## 4 RELATED WORK

Various results have been established in RNNs approximation theory, see Sontag (1998); Hanson et al. (2021) and references therein. For unbounded input index sets, $L_p$ approximation is established by Gonon & Ortega (2020). In Gonon & Ortega (2021), the universal approximation theorem is constructed for functionals with fading memory in the discrete time setting. In Li et al. (2020), the universal approximation theorem and Jackson-type approximation theorem characterize the density and speed of linear RNNs applied to linear functionals. Most existing results are forward (Jackson-type) approximation theorems, which upper bound the optimal approximation error. Of most relevance is the Bernstein-type result proved in Li et al. (2022), where it has been proved that the linear functional sequences that can be efficiently approximated by linear RNNs must have an exponential decaying memory. However, the main limitation of the above result is the linear setting for both models and targets.

The notion of approximation stability is one of the central concepts we exploit in this paper. We note that in classical approximation theory, stable approximation has numerous definitions depending on the setting (DeVore et al., 2021). For example, in nonlinear approximation (DeVore, 1998), a stably approximating sequence $\{H_m\}$ of $H$ is one that satisfies $|H_m| \leq C|H|$ for some absolute constant $C > 0$ and all $m$. This approach is taken to show the non-existence of stable procedure to approximating functions from equally-spaced samples with exponential convergence on analytic functions (Platte et al., 2011). This notion of stability is on the conditioning of the approximation problem. In contrast, our notion of stability introduced in Section 3.2 is more similar to a uniform continuity requirement. Pertaining to sequence modeling, a related but different notion of dynamic stability (Hanson & Raginsky, 2020) was used to prove a Jackson-type results for universal simulation of dynamical systems. There, the stability is akin to requiring the uniform (in inputs) continuity of the flow-map of the RNN hidden dynamics. In practice, some specific forms of the stable reparameterization we defined in Equation (16) has been adopted in state-space models optimization (Gu et al., 2020; 2021; Smith et al., 2023; Wang & Xue, 2023; Wang & Li, 2023).

## 5 CONCLUSION

In summary, we derive an inverse approximation result in the setting of sequence modeling using nonlinear RNNs. We show that, assuming that a given target sequence relationship (mathematically understood as a nonlinear functional sequence) can be stably approximated by RNNs with nonlinear activations, then the target functional sequence's memory structure must be exponentially decreasing. This places a priori limitations on the ability of RNNs to learn long-term memory in sequence modeling, and makes precise the empirical observation that RNNs do not perform well for such problems. From the approximation viewpoint, our results show that this failure is not only due to learning algorithms (e.g. explosion of gradients), but also due to fundamental limitations of the RNN hypothesis space. At the same time, our analysis points to reparameterization as a principled methodology to remedy the limitations of RNN when it comes to long-term memory and we demonstrate its effectiveness in by learning linear functionals with polynomial memory.

ACKNOWLEDGMENTS

This research is supported by the National Research Foundation, Singapore, under the NRF fellowship (project No. NRF-NRFF13-2021-0005). Shida Wang is supported by NUS-RMI Scholarship.

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

# A  THEORETICAL RESULTS AND PROOFS

In this section, we first attach the famous Riesz representation theorem and the previous results for linear RNNs in Section A.2 and Section A.3.

## A.1  WELL-DEFINEDNESS OF CONTINUOUS-TIME RNN

Here we discuss the well-definedness of continuous-time nonlinear RNNs. The $-\infty$ boundary condition is chosen to make the time-homogeneity well-defined, representing a more general scenario than the boundary at 0. Regarding issues on the input continuity, Heaviside inputs are point-wise limit of smooth inputs $\mathbf{u}^x = \lim_{k\to\infty} \mathbf{u}^x_{k,cont}$. Therefore the outputs of continuous targets functionals are well-defined: $H_t(\mathbf{u}^x) = H_t(\lim_{k\to\infty} \mathbf{u}^x_{k,cont}) = \lim_{k\to\infty} H_t(\mathbf{u}^x_{k,cont})$.

## A.2  RIESZ REPRESENTATION THEOREM

**Theorem A.1** (Riesz-Markov-Kakutani representation theorem). *Assume* $H : C_0(\mathbb{R}^d) \mapsto \mathbb{R}$ *is a linear and continuous functional. Then there exists a unique, vector-valued, regular, countably additive signed measure* $\mu$ *on* $\mathbb{R}$ *such that*

$$H(\mathbf{x}) = \int_{\mathbb{R}} x_s^\top d\mu(s) = \sum_{i=1}^d \int_{\mathbb{R}} x_{s,i} d\mu_i(s). \tag{17}$$

*In addition, we have* $\|H\| := \sup_{\|\mathbf{x}\|_{\mathcal{X}} \leq 1} |H(\mathbf{x})| = \|\mu\|_1(\mathbb{R}) := \sum_i |\mu_i|(\mathbb{R})$.

Based on the representation theorem, one can further obtain that the functionals $\{H_t : t \in \mathbb{R}\}$ satisfying properties in Definition 2.1 can be represented by the following convolutional form

$$H_t(\mathbf{x}) = \int_0^\infty x_{t-s}^\top \rho(s) ds, \quad t \in \mathbb{R}, \tag{18}$$

and the representation function $\rho : [0, \infty) \to \mathbb{R}^d$ is a measurable and integrable function with $\|\rho\|_{L^1([0,\infty))} = \sup_{t\in\mathbb{R}} \|H_t\|$. The details can be found in Appendix A.3 or Li et al. (2020), where the causality and time-homogeneity play a key role.

## A.3  THREE TYPES OF APPROXIMATION FOR LINEAR FUNCTIONALS AND LINEAR RNNS

We include the statements of the universal approximation theorem and approximation rate for linear functionals by linear RNNs from Li et al. (2020).

**Theorem A.2** (Universal approximation for linear functionals by linear RNNs (Li et al., 2020)). *Let* $\{H_t : t \in \mathbb{R}\}$ *be a sequence of linear, continuous, causal, regular and time-homogeneous functionals defined on* $\mathcal{X}$. *Then, for any* $\epsilon > 0$, *there exists* $\{\bar{H}_t : t \in \mathbb{R}\} \in \bar{\mathcal{H}}^{\text{linear}}$ *such that*

$$\sup_{t\in\mathbb{R}} \|H_t - \bar{H}_t\| \equiv \sup_{t\in\mathbb{R}} \sup_{\|\mathbf{x}\|_{\mathcal{X}} \leq 1} |H_t(\mathbf{x}) - \bar{H}_t(\mathbf{x})| \leq \epsilon. \tag{19}$$

**Theorem A.3** (Jackson-type approximation rate for linear functionals by linear RNNs (Li et al., 2020)). *Assume the same conditions as Theorem A.2. Consider the output of constant signals*

$$y_i(t) = H_t(\mathbf{e}_i), \quad i = 1, \dots, d,$$

*where* $\mathbf{e}_i$ *is a constant signal with* $e_{i,t} = e_i \mathbf{1}_{\{t\geq 0\}}$, *and* $\{e_i\}_{i=1}^d$ *denote the standard basis vectors in* $\mathbb{R}^d$. *Suppose there exist constants* $\alpha \in \mathbb{N}_+, \beta, \gamma > 0$ *such that for* $i = 1, \dots, d$, $k = 1, \dots, \alpha + 1$, $y_i(t) \in C^{(\alpha+1)}(\mathbb{R})$ *and*

$$e^{\beta t} y_i^{(k)}(t) = o(1), \quad as \ t \to +\infty,$$

$$\sup_{t\geq 0} \frac{|e^{\beta t} y_i^{(k)}(t)|}{\beta^k} \leq \gamma.$$

*Then, there exists a universal constant* $C(\alpha)$ *only depending on* $\alpha$, *such that for any* $m \in \mathbb{N}_+$, *there exists a sequence of width-$m$ RNN functionals* $\{\bar{H}_t : t \in \mathbb{R}\} \in \bar{\mathcal{H}}_m^{\text{linear}}$ *such that*

$$R(m) = \sup_{t\in\mathbb{R}} \|H_t - \bar{H}_t\| \equiv \sup_{t\in\mathbb{R}} \sup_{\|\mathbf{x}\|_{\mathcal{X}} \leq 1} |H_t(\mathbf{x}) - \bar{H}_t(\mathbf{x})| \leq \frac{C(\alpha)\gamma d}{\beta m^\alpha}. \tag{20}$$

**Theorem A.4** (Bernstein-type approximation for linear functionals by linear RNNs (Li et al., 2022)).
*Let $\{H_t : t \in \mathbb{R}\}$ be a sequence of linear, continuous, causal, regular and time-homogeneous functionals defined on $\mathcal{X}$. Consider the output of constant signals*

$$y_i(t) = H_t(\mathbf{e}_i) \in C^{(\alpha+1)}(\mathbb{R}), \quad i \in 1, \ldots, d, \; \alpha \in \mathbb{N}_+. \tag{21}$$

*Suppose that for each $m \in \mathbb{N}_+$, there exists a sequence of width-$m$ RNN functionals $\{\bar{H}_t : t \in \mathbb{R}\} \in \bar{\mathcal{H}}_m^{\text{linear}}$ approximating $H_t$ in the following sense:*

$$\lim_{m \to \infty} \sup_{t \geq 0} |\bar{y}_{i,m}^{(k)}(t) - y_i^{(k)}(t)| = 0, \quad i = 1, \ldots, d, \; k = 1, \ldots, \alpha + 1, \tag{22}$$

*where*

$$\bar{y}_{i,m}(t) = \bar{H}_t(\mathbf{e}_i), \quad i = 1, \ldots, d. \tag{23}$$

*Define $w_m = \max_{j \in [m]} Re(\lambda_j)$, where $\{\lambda_j\}_{j=1}^m$ are the eigenvalues of $\bar{W}$ in $\{\bar{H}_t : t \in \mathbb{R}\}$. Assume the parameters are uniformly bounded and there exists a constant $\beta > 0$ such that $\limsup_{m \to \infty} w_m < -\beta$, then we have*

$$e^{\beta t} y_i^{(k)}(t) = o(1) \text{ as } t \to +\infty, \quad i = 1, \ldots, d, \; k = 1, \ldots, \alpha + 1. \tag{24}$$

## A.4 WHY SOLVE CURSE OF MEMORY BY ADDING NONLINEARITY?

In feed-forward neural networks, the inclusion of nonlinear activation is known to fundamentally change the approximation capacities of multi-layer networks. Without this nonlinear activation, the model simply performs a linear transformation. In sequence modeling, linear RNNs serve as universal approximators for linear functionals. However, they struggle to learn linear functionals with polynomial decaying memory. Given this context, one might naturally wonder if incorporating nonlinear activation could expand the hypothesis space in such a way as to overcome this 'curse of memory' problem.

Section 3.3 indicates that the presence of nonlinear activation does not fundamentally alter the memory pattern of models. Consequently, for stable approximation, targets must demonstrate an exponentially decaying memory. The proofs also show that hardtanh/tanh RNNs require the eigenvalues to be bounded away from 0.

The findings in Section 3.3 lead to the proposal of stable parameterisation as a potential solution for long-memory learning. Stable parameterisation ensures that the real parts of eigenvalues for the recurrent weight matrix remain negative, even when the weights undergo perturbations. In Figure 4, we demonstrate that linear RNNs can stably approximate linear functionals with polynomial decay memory when the recurrent weight matrix is parameterised using exponential/softplus parameterisation. Moreover, by selecting linear functionals with either exponential or polynomial decay memory, we can clearly change the targets' memory structures.

## A.5 POINT-WISE CONTINUITY LEADS TO DECAYING MEMORY

Here we give the proof of decaying memory based on the point-wise continuity of $\frac{dH_t}{dt}$ and boundedness and time-homogeneity of $\mathbf{H}$:

*Proof.*

$$\lim_{t \to \infty} \left| \frac{dH_t}{dt}(\mathbf{u}^x) \right| = \lim_{t \to \infty} \left| \frac{dH_0}{dt}(x \cdot \mathbf{1}_{\{s \geq -t\}}) \right| = \left| \frac{dH_0}{dt}(\mathbf{x}) \right| = 0.$$

The first equation comes from time-homogeneity. The second equation is derived from the point-wise continuity where input $\mathbf{x}$ means constant $x$ for all time $\mathbf{x} = x \cdot \mathbf{1}_{\{s \geq -\infty\}}$. By time-homogeneity and boundedness, the output over constant input is the same: $H_t(\mathbf{x}) = H_s(\mathbf{x})$ for all $s, t$. Therefore $|\frac{dH_0}{dt}(\mathbf{x})| = 0$. □

## A.6 POINT-WISE CONTINUITY OF NONLINEAR RNNS

We prove that nonlinear RNNs with Lipschitz continuous activations are point-wise continuous over bounded, piece-wise constant input sequences, including the Heaviside inputs.

*Proof.* Without loss of generality, assume $t > 0$. The following $|\cdot|$ refers to $p = \infty$ norm.

Assume the continuous inputs sequence converges to some bounded piece-wise constant input sequence $\mathbf{x}$ in point-wise sense: $\lim_{k \to \infty} \mathbf{x_k} = \mathbf{x}$. For any bounded interval $[-B, B]$, we can see the outputs are well-defined and the point-wise (in $t$) convergence holds:

Let $h_{k,t}$ and $h_t$ be the hidden states over $\mathbf{x_k}$ and $\mathbf{x}$. By definition and triangular inequality

$$\frac{d|h_{k,t} - h_t|}{dt} = |\sigma(Wh_{k,t} + Ux_{k,t}) - \sigma(Wh_t + Ux_t)| \leq L(|W||h_{k,t} - h_t| + |U||x_{k,t} - x_t|).$$

Here $L$ is the Lipschitz constant of activation $\sigma$.

Apply Grönwall inequality to the above inequality, we have:

$$|h_{k,t} - h_t| \leq \int_0^t e^{L|W|(t-s)} L|U||x_{k,s} - x_s|ds$$

As the inputs are bounded, by dominated convergence theorem we have RHS converges to 0 therefore

$$\lim_{k \to \infty} |h_{k,t} - h_t| = 0.$$

Therefore the point-wise convergence of $\frac{dH_t}{dt}$ can be achieved:

$$\lim_{k \to \infty} |\frac{dy_{k,t}}{dt} - \frac{dy_t}{dt}| \leq \lim_{k \to \infty} |c|L(|W||h_{k,t} - h_t| + |U||x_{k,t} - x_t|) = 0.$$

So we have: $\lim_{k \to \infty} \frac{dy_{k,t}}{dt} = \frac{dy_t}{dt}$. Therefore $\frac{dH_t}{dt}$ are point-wise continuous functionals over "bounded piece-wise constant input sequence". $\qquad \square$

### A.7 COMPARISON OF DECAYING MEMORY AND FADING MEMORY

The targets with decaying memory is a common assumption in the sense that it can also be derived from the uniformly asymptotically incrementally stable assumption from Hanson & Raginsky (2020), which is demonstrated by 'imperfect system models may still be capable of generating outputs that uniformly approximate the outputs of the original system over infinite time intervals'.

There is a concept of **fading memory** introduced in the nonlinear functional approximation with Volterra series (Boyd et al., 1984). A functional is said to have fading memory if there exists a monotonically decreasing function $w : \mathbb{R}_+ \to (0, 1]$, $\lim_{t \to \infty} w(t) = 0$, such that for any $\epsilon > 0$, there exists a $\delta > 0$ such that for any $\mathbf{x}, \mathbf{x}' \in \mathcal{X}$,

$$|H_t(\mathbf{x}) - H_t(\mathbf{x}')| < \epsilon \text{ whenever } \sup_{s \in (-\infty, t]} |\mathbf{x}_s - \mathbf{x}'_s| w(t - s) < \delta. \tag{25}$$

*Remark* A.5. In terms of the relations between fading memory and decaying memory, they are independent. On one hand, there exists a so-called peak-hold operator (Boyd et al., 1984) with a decaying memory (but not fading memory). On the other hand, it is possible for a linear functional to only have the fading memory (but no decaying memory), where it needs only the requirement $\int_0^\infty \frac{|\rho(s)|_2}{|w(s)|} ds < \infty$, which does not imply $\lim_{s \to \infty} |\rho(s)|_2 = 0$.

### A.8 INVERSE APPROXIMATION THEOREM FOR LINEAR-LIKE ACTIVATIONS

We hereafter call $\widetilde{\mathbf{H}}_m = \widehat{\mathbf{H}}(\cdot, \tilde{\theta}_m)$ to be the perturbed models.

We will briefly summarize the idea of the proof. Given that approximations are defined to be stable, the decaying memory property ensures that the derivative of the hidden states for the perturbed model approaches 0 as time $t \to \infty$. The decay rate of the memory function is characterised by the rate at which $\frac{d\tilde{h}_t}{dt}$ converges to 0. Using the Hartman-Grobman theorem, we can obtain bounds on the eigenvalues of the matrices $W_m$. These bounds, in turn, determine the decay rate of $\frac{d\tilde{h}_t}{dt}$, leading to an exponential decay in the model's memory function. Finally, since the models with uniformly

exponential decaying memory can only approximate targets with exponential decay, the memory function of the nonlinear target functionals must also be decaying exponentially.

For simplicity of the notation, we strengthen the assumption from $\limsup_{m\to\infty}$ to $\lim_{m\to\infty}$. To get the result for $\limsup_{m\to\infty}$, we only need to consider the subsequence of $m_k$ with a limit.

*Proof.* Define the derivative of hidden states (for unperturbed model $\widehat{H}_t(\cdot,\theta_m)$) to be $v_{m,t} = \frac{dh_{m,t}}{dt}$, similarly, $\tilde{v}_{m,t}$ is the derivative of hidden states for perturbed models ($\widetilde{H}_t = \widehat{H}_t(\cdot,\tilde{\theta}_m)$), $\tilde{v}_{m,t} = \frac{d\tilde{h}_{m,t}}{dt}$.

Since each perturbed model has a decaying memory, by Lemma A.6, we have

$$\lim_{t\to\infty} \tilde{v}_{m,t} = 0, \quad \forall m. \tag{26}$$

In particular, for each $m$ and perturbation within the stability radius, there exists $t_0$ which depends on $m$ and $Z_0$ such that $|\tilde{v}_{m,t}|_\infty < Z_0, t \geq t_0$.

If the inputs are limited to Heaviside input, the derivative of perturbed models' hidden states $\tilde{v}_{m,t}$ satisfies the following dynamics:

$$\frac{d\tilde{v}_{m,t}}{dt} = \sigma'(\tilde{v}_{m,t}) \circ (\widetilde{W}_m \tilde{v}_{m,t}) = c_\sigma \widetilde{W}_m \tilde{v}_{m,t}, \quad t \geq t_0 \tag{27}$$

$$\tilde{v}_{m,t_0} = \sigma(\widetilde{W}_m h_{t_0} + \widetilde{U}_m x_{t_0} + \tilde{b}_m), \quad |\tilde{v}_{m,t_0}|_\infty < Z_0. \tag{28}$$

Select a sequence of perturbed recurrent matrices $\{\widetilde{W}_{m,k}\}_{k=1}^\infty$ satisfying the following two properties:

1. $c_\sigma \widetilde{W}_{m,k}$ is Hyperbolic, which means the real part of the eigenvalues of the matrix are nonzero.

2. $\lim_{k\to\infty}(\widetilde{W}_{m,k} - W_m) = \beta_0 I_m$. (Notice that $|\widetilde{W}_{m,k} - W_m|_2 \leq \beta_0 I_m$ for all $k$)

Moreover, by Lemma A.7, we know the each hyperbolic matrix $c_\sigma \widetilde{W}_{m,k}$ is Hurwitz as the system for $\tilde{v}_{m,t}$ is asymptotically stable.

$$\max_{i\in[m]}(\text{Re}(\lambda_i(\widetilde{W}_m))) < 0, \tag{29}$$

$$\max_{i\in[m]}(\text{Re}(\lambda_i(W_m))) < -\beta_0. \tag{30}$$

Therefore the original unperturbed recurrent weight matrix $W_m$ satisfies the following eigenvalue inequality **uniformly** in $m$.

$$\sup_m \max_{i\in[m]}(\text{Re}(\lambda_i(W_m))) < -\beta_0. \tag{31}$$

As the memory functions of the perturbed models are uniformly decaying, we consider a subclass of perturbed models where only the weight of linear readout map $c_m$ are perturbed. In other words, we are considering perturbed models with weights $(W_m, U_m, b_m, \tilde{c}_m)$. Since the supremum of perturbed model memory functions are decaying:

$$\lim_{t\to\infty} \sup_m \left|\frac{d}{dt}\tilde{y}_{m,t}\right| = 0. \tag{32}$$

For any $\epsilon > 0$, there exists a $T_\epsilon > 0$ such that for all $t > T_\epsilon$,

$$\sup_{|\tilde{c}_m - c_m|_2 \leq \beta_0} \left|\frac{d}{dt}\tilde{y}_{m,t}\right| = \sup_{|\tilde{c}_m - c_m|_2 \leq \beta_0} |\tilde{c}_m^\top v_{m,t}| < \epsilon. \tag{33}$$

We have

$$|c_m^\top v_{m,t}| + \beta_0 |v_{m,t}|_2 < \epsilon, \tag{34}$$

$$|v_{m,t}|_2 < \frac{\epsilon}{\beta_0}, \qquad t > T_\epsilon. \tag{35}$$

Select $\epsilon = \frac{1}{2}\beta_0 Z_0$, we have the **unperturbed** derivative $|v_{m,t}|_2 < Z_0$ for $t > T_\epsilon$. Therefore the dynamics of $v_{m,t}$ is a linear system for $t > T_\epsilon$. For $t \geq T_\epsilon$

$$|v_{m,t}|_2 = |e^{W_m(t-T_\epsilon)}v_{m,T_\epsilon}|_2 \tag{36}$$

$$\leq |e^{W_m(t-T_\epsilon)}|_2 |v_{m,T_\epsilon}|_2 \tag{37}$$

$$\leq e^{-\beta_0(t-T_\epsilon)}|v_{m,T_\epsilon}|_2 \tag{38}$$

The model memory decays exponentially: For $t \geq T_\epsilon$,

$$|c_m^\top v_{m,t}| \leq |c_m|_2 |v_{m,T_\epsilon}|_2 e^{-\beta_0(t-T_\epsilon)} \tag{39}$$

$$\leq |c_m|_2 Z_0 e^{-\beta_0(t-T_\epsilon)} \tag{40}$$

$$= |c_m|_2 Z_0 e^{\beta_0 T_\epsilon} e^{-\beta_0 t} \tag{41}$$

For $t \in [0, T_\epsilon]$, as the memory function is continuous, there exists a constant $C' := \sup_{t \in [0, T_\epsilon]} e^{\beta_0 t}|c_m^\top v_{m,t}|$ such that

$$|c_m^\top v_{m,t}| \leq C' e^{-\beta_0 t}, \quad t \in [0, T_\epsilon] \tag{42}$$

There exists constant $C_0 := \max(\sup_m |c_m|_2 \cdot Z_0 e^{\beta_0 T_\epsilon}, C')$ such that for any $m$,

$$\mathcal{M}(\widehat{\mathbf{H}}_m)(t) = \sup_{x \neq 0} \frac{1}{|x|_\infty + 1}\left|\frac{d}{dt}y_{m,t}\right| \leq |c_m^\top v_{m,t}| \leq C_0 e^{-\beta_0 t}, \quad t \geq 0. \tag{43}$$

Last, by Lemma A.10, the target $\mathbf{H}$ has an exponentially decaying memory as it is approximated by a sequence of models $\{\widehat{\mathbf{H}}_m\}_{m=1}^\infty$ with uniformly exponentially decaying memory. $\qquad\square$

## A.9 INVERSE APPROXIMATION THEOREM FOR TANH-LIKE ACTIVATIONS

Next we give the proof for RNNs using tanh-like activations.

*Proof.* Similar to the previous proof, we still have

$$\lim_{t \to \infty} \tilde{v}_{m,t} = 0, \quad \forall m. \tag{44}$$

In particular, for each $m$ and perturbation within the stability radius, there exists $t_0$ such that $|\tilde{v}_{m,t}|_\infty < Z_0, t \geq t_0$.

Now, the perturbed hidden states satisfies the following dynamics:

$$\frac{d\tilde{v}_{m,t}}{dt} = \sigma'_{\tanh}(\tilde{v}_{m,t}) \circ \widetilde{W}_m \tilde{v}_{m,t} = (I - \text{Diag}(\tilde{v}_{m,t})^2)\widetilde{W}_m \tilde{v}_{m,t}, \tag{45}$$

$$\tilde{v}_{m,0} = \sigma_{\tanh}(\widetilde{U}_m x_0 + \tilde{b}_m). \tag{46}$$

Select a sequence of perturbed recurrent matrices $\{\widetilde{W}_{m,k}\}_{k=1}^\infty$ satisfying the following two properties:

1. $\widetilde{W}_{m,k}$ is Hyperbolic, which means the real part of the eigenvalues of the matrix are nonzero.

2. $\lim_{k \to \infty}(\widetilde{W}_{m,k} - W_m) = \beta_0 I_m$.

By Lemma A.7, we know each hyperbolic matrix $\widetilde{W}_{m,k}$ is Hurwitz as the target functional sequence has a stable approximation. Similarly, we have the following uniform bound on eigenvalues of $\{W_m\}$:

$$\sup_m \max_{i \in [m]}(\text{Re}(\lambda_i(W_m))) \leq -\beta_0. \tag{47}$$

Since every $W_m$ is Hurwitz matrix, consider the corresponding (continuous) Lyapunov equation (Khalil, 2002)

$$W_m^T P_m + P_m W_m = -Q_m. \tag{48}$$

For simplicity, we select the matrix $Q_m = I_m$. For any positive definite matrix $Q_m$, it is known that $P_m$ has an explicit integral form:

$$P_m = \int_0^\infty e^{W_m^\top t} Q_m e^{W_m t} dt = \int_0^\infty e^{W_m^\top t} I_m e^{W_m t} dt. \tag{49}$$

By Lemma A.8, we know $P_m$ has a uniformly bounded $L_2$ norm

$$\sup_m \|P_m\|_2 \le \frac{1}{2\beta_0}. \tag{50}$$

We construct a Lyapunov function $V(v) = v^T P_m v \le \frac{1}{2\beta_0} |v|_2^2$, which satisfies the following differential equation:

$$
\begin{aligned}
\frac{dV(v_{m,t})}{dt} =& v_{m,t}^\top (W_m^\top (I - \mathrm{Diag}(v_{m,t})^2) P_m + P_m (I - \mathrm{Diag}(v_{m,t})^2) W_m) v_{m,t} \\
=& v_{m,t}^\top (W_m^\top P_m + P_m W_m) v_{m,t} \\
& - v_{m,t}^\top (W_m^\top \mathrm{Diag}(v_{m,t})^2 P_m + P_m \mathrm{Diag}(v_{m,t})^2 W_m) v_{m,t} \\
=& - |v_{m,t}|_2^2 - v_{m,t}^\top (W_m^\top \mathrm{Diag}(v_{m,t})^2 P_m + P_m \mathrm{Diag}(v_{m,t})^2 W_m) v_{m,t}.
\end{aligned}
\tag{51}
$$

By Lemma A.9, for any positive $L \in (0,1)$, there is an $\Upsilon_L = \sqrt{\frac{\beta_0 L}{M_0}} > 0$ such that the following inequality holds for any $m$,

$$|v_{m,t}^\top (W_m^\top \mathrm{Diag}(v_{m,t})^2 P_m + P_m \mathrm{Diag}(v_{m,t})^2 W_m) v_{m,t}| \le L |v_{m,t}|_2^2, \quad \forall |v_{m,t}|_2 \le \Upsilon_L. \tag{52}$$

As the memory functions of the perturbed models are uniformly decaying, we consider a subclass of perturbed models where only the linear readout map $c_m$ are perturbed. Since the limit of perturbed model memory functions are decaying:

$$\lim_{t \to \infty} \sup_m \left| \frac{d}{dt} \tilde{y}_{m,t} \right| = 0. \tag{53}$$

For any $\epsilon > 0$, there exists a $T_\epsilon > 0$ such that for all $t > T_\epsilon$

$$\sup_{|\tilde{c}_m - c_m|_2 \le \beta_0} \left| \frac{d}{dt} \tilde{y}_{m,t} \right| = \sup_{|\tilde{c}_m - c_m|_2 \le \beta_0} |\tilde{c}_m^\top v_{m,t}| < \epsilon. \tag{54}$$

We have

$$|c_m v_{m,t}| + \beta_0 |v_{m,t}|_2 < \epsilon, \tag{55}$$

$$|v_{m,t}|_2 < \frac{\epsilon}{\beta_0}. \tag{56}$$

Select $\epsilon < \beta_0 \Upsilon_L$, we have

$$|v_{m,t}|_2 < \frac{\epsilon}{\beta_0} < \Upsilon_L. \tag{57}$$

The Lyapunov function satisfies the following inequality for $t \ge T_\epsilon$

$$
\begin{aligned}
\frac{dV(v_{m,t})}{dt} =& - |v_{m,t}|_2^2 - v_{m,t}^\top (W_m^\top \mathrm{Diag}(v_{m,t})^2 P_m + P_m \mathrm{Diag}(v_{m,t})^2 W_m) v_{m,t} & (58) \\
\le& - |v_{m,t}|_2^2 + L |v_{m,t}|_2^2 & (59) \\
=& - (1 - L) |v_{m,t}|_2^2 & (60) \\
\le& - 2(1 - L) \beta_0 V(v_{m,t}). & (61)
\end{aligned}
$$

The Lyapunov function $V(v_{m,t})$ is decaying exponentially for $\epsilon < \beta_0 \Upsilon_L, t \ge T_\epsilon$,

$$V(v_{m,t}) \le e^{-2(1-L)\beta_0 (t - T_\epsilon)} V(v_{m,T_\epsilon}). \tag{62}$$

Notice that $|v_{m,t}|_2 < \frac{\epsilon}{\beta_0} < \Upsilon_L$. Therefore

$$V(v_{m,t}) \le e^{-2(1-L)\beta_0 (t - T_\epsilon)} \cdot \frac{1}{2\beta_0} \Upsilon_L^2. \tag{63}$$

Since the model weights are uniformly bounded, there is a constant $M$ such that

$$\sup_m |W_m|_2 = M < \infty. \tag{64}$$

Since $|I_m|_2 = 1 \leq 2|W_m|_2|P_m|_2$, this implies

$$\min_m |P_m|_2 \geq \frac{1}{2M} > 0. \tag{65}$$

As $V(v_{m,t}) \geq \frac{1}{2M}|v_{m,t}|_2^2$, there exists a uniform constant $C_0 := \sqrt{\frac{M}{\beta_0}}\Upsilon_L = \sqrt{\frac{ML}{M_0}}$ such that

$$|v_{m,t}|_2 \leq C_0 e^{-(1-L)\beta_0(t-T_\epsilon)}, \quad \forall t > T_\epsilon. \tag{66}$$

Since the model weights $c_m$ are uniformly (in $m$) bounded $C_1 := \sup_m |c_m|_2 < \infty$. Therefore the model memories are uniformly (in $m$) exponentially decaying

$$\sup_m \mathcal{M}(\widehat{\mathbf{H}}_m)(t) = \sup_m \sup_{x \neq 0} \frac{\left|\frac{d}{dt}\widehat{H}_{m,t}(\mathbf{u}^x)\right|}{|x|_\infty + 1} \leq C_0 C_1 e^{-(1-L)\beta_0(t-T_\epsilon)}. \tag{67}$$

Notice that $T_\epsilon$ is independent of $m$, it only depends on $\epsilon$.

Take $L \to 0$, for any $\beta < \beta_0$ there exists a constant $C^*$ which depends on $\beta$ such that

$$\sup_m \mathcal{M}(\widehat{\mathbf{H}}_m)(t) \leq C^* e^{-\beta t}. \tag{68}$$

Last, by Lemma A.10, the target has exponentially decaying memory as it is approximated by a sequence of models with uniformly exponentially decaying memory. □

## A.10 PROOFS OF LEMMAS

In the following section we include the lemmas used in the proof of main theorems. Lemma A.10 is used in the proof for Theorem 3.9. Lemmas A.6 to A.10 are applied in Appendix A.9.

*Lemma A.6. Assume the target functional sequence has a $\beta_0$-stable approximation and the perturbed model has a decaying memory, we show that $\tilde{v}_{m,t} \to 0$ for all $m$.*

*Proof.* For any $m$, fix $\widetilde{W}_m$ and $\widetilde{U}_m$. Since the perturbed model has a decaying memory,

$$\lim_{t \to \infty} \left|\frac{d}{dt}\widetilde{H}_m(\mathbf{u}^x)\right| = \lim_{t \to \infty} |\tilde{c}_m^\top \tilde{v}_{m,t}| = 0. \tag{69}$$

According to the definition of stable approximation about perturbation, as $t \to \infty$, $\tilde{v}_{m,t}$ vanishes over the projection to $\tilde{c}_m$ for any $|\tilde{c}_m - c_m|_\infty \leq \beta$. By linear algebra, there exist $\{\Delta c_i\}_{i=1}^m$, $|\Delta c_i|_\infty < \beta$ such that $c_m + \Delta c_1, \ldots, c_m + \Delta c_m$ form a basis of $\mathbb{R}^m$. We can then decompose any vector $u$ into

$$u = k_1(c_m + \Delta c_1) + \cdots + k_m(c_m + \Delta c_m). \tag{70}$$

Take the inner product of $u$ and $\tilde{v}_{m,t}$, we have

$$\lim_{t \to \infty} u^\top \tilde{v}_t = \sum_{i=1}^m k_i \lim_{t \to \infty} (c_m + \Delta c_i)^\top \tilde{v}_t = 0 \tag{71}$$

As the above result holds for any vector $u$, we get

$$\lim_{t \to \infty} \tilde{v}_{m,t} = 0. \tag{72}$$

□

*Lemma A.7. Consider a dynamical system with the following dynamics:*

$$\frac{dv_t}{dt} = Diag(\sigma'(v_t))Wv_t,$$
$$v_0 = \sigma(Ux_0 + b). \tag{73}$$

*If $W \in \mathbb{R}^{m \times m}$ is hyperbolic and the system in Equation (73) is asymptotically stable over any bounded Heaviside input $|x_0|_1 < B$, then the matrix $W$ is Hurwitz.*

*Proof.* When $\sigma$ is the linear-like activation, $\sigma'(z) = c_\sigma$ for $|z| \leq 1$, $\mathrm{Diag}(\sigma'(v_{m,t})) = c_\sigma I_m$. When $\sigma$ is the tanh-like activation, $\sigma'(z) = a - b\sigma(z)^2$. Also, $\sigma^{-1}$ is continuous at 0, therefore $\sigma$ is an open mapping around 0. For sufficiently large $B$, there exists $\delta_m > 0$ such that a small ball centered at 0 is contained in the initializations that are stable for the system in Equation (73).

$$v_0 \in B(0, \delta_m) \subseteq \{\sigma(Ux_0 + b) : |x_0|_1 < B\}. \tag{74}$$

For systems with linear-like activations, since $\lim_{t \to \infty} v_t = 0$ for any $v_0 \in B(0, \delta_m)$, it implies the local asymptotic stability at the origin. If $W$ has an eigenvalue with a positive real part, according to Hartman-Grobman theorem, $\tilde{v}_t$ has an unstable manifold locally at the origin. However, this contradicts the asymptotic stability around the origin with the initialization $v_0$ in $B(0, \delta_m) \subset \mathbb{R}^m$.

For systems with tanh-like activations, again by Hartman-Grobman theorem, the asymptotic stability is determined by the linearized system, therefore $W$ is Hurwitz. $\square$

*Lemma A.8. Given a sequence of Lyapunov equation with Hurwitz matrices $\{W_m\}_{m=1}^\infty$*

$$W_m^\top P_m + P_m W_m = -I_m. \tag{75}$$

*Assume the eigenvalues of $W_m$ are uniformly bounded away from 0:*

$$\sup_{i \in [m]} Re(\lambda_i(W_m)) \leq -\beta_0. \tag{76}$$

*Show that the $L_2$ norm of $P_m$ are uniformly bounded*

$$\sup_m \|P_m\|_2 \leq \frac{1}{2\beta_0}. \tag{77}$$

*Proof.* By the theory of Lyapunov equation (Khalil, 2002, p. 136), it can be verified that the symmetric positive-definite matrix

$$P_m = \int_0^\infty e^{W_m^\top t} I_m e^{W_m t} dt \tag{78}$$

is the solution to the above Lyapunov equation.

Assume $W_m$'s eigenvalue-eigenvector couples are $(\lambda_i, v_i), i \in [m]$. Without loss of generality, we assume $v_i$ are all unit eigenvectors.

$$\|v_i\|_2 = 1, \quad 1 \leq i \leq m. \tag{79}$$

For simplicity, we first assume the eigenvalues are distinct. Therefore, any unit vector $u$ can be decomposed into linear combination of eigenvectors

$$u = \sum_{i=1}^m c_i v_i, \quad \sum_{i=1}^m c_i^2 = 1. \tag{80}$$

We have

$$u^\top P_m u = \int_0^\infty \|e^{W_m t} u\|_2^2 dt \tag{81}$$

$$= \int_0^\infty \sum_{i=1}^m c_i^2 e^{2\lambda_i t} \|v_i\|_2^2 dt \tag{82}$$

$$= \int_0^\infty \sum_{i=1}^m c_i^2 e^{2\lambda_i t} dt \tag{83}$$

$$\leq \int_0^\infty \sum_{i=1}^m c_i^2 e^{-2\beta_0 t} dt \tag{84}$$

$$= \int_0^\infty e^{-2\beta_0 t} dt = \frac{1}{2\beta_0}. \tag{85}$$

Therefore, $\|P_m\|_2 \leq \frac{1}{2\beta_0}$. Notice the bound on $L_2$ norm is uniform as the eigenvalue bound on $W_m$ is uniform.

Next, if $W_m$ has repeated eigenvalues, we can consider $W_m$ with slight perturbations such that the perturbed $\widetilde{W}_m$ are diagonalizable. In this case the eigenvalues might not be distinct but it's feasible to construct the eigenvectors to be orthogonal to each other. We can bound the perturbed matrix's $L_2$ norm with the previous argument. The corresponding $\widetilde{P}_m$ converges to $P_m$ by the continuity of explicit form in Equation (78). $\qquad\square$

*Lemma A.9. Assume $\{W_m \in \mathbb{R}^{m \times m}\}_{m=1}^{\infty}$ is a sequence of Hurwitz matrices with eigenvalues bounded away from 0:*

$$\sup_m \max_{i \in [m]} (Re(\lambda_i(W_m))) \leq -\beta_0. \tag{86}$$

*Assume the max norms for matrices $\{W_m\}_{m=1}^{\infty}$ are uniformly bounded:*

$$\sup_m \sup_{1 \leq i,j \leq m} |W_{m,ij}| \leq M_0. \tag{87}$$

*Define the solution to the following Lyapunov equation to be $P_m$:*

$$W_m^\top P_m + P_m W_m = -I_m. \tag{88}$$

*We show that for any positive constant $L$, there exists an $\Upsilon > 0$ such that the following inequality holds for any $m$ and $|v_m|_2 \leq \Upsilon$:*

$$|v_m^\top (W_m^\top Diag(v_m)^2 P_m + P_m Diag(v_m)^2 W_m) v_m| \leq L |v_m|_2^2. \tag{89}$$

*Proof.* First, by the property of matrix and vector norm:

$$|v_m^\top P_m \mathrm{Diag}(v_m)^2 W_m v_m| \leq |v_m|_2 |P_m|_2 |\mathrm{Diag}(v_m)^2 W_m v_m| \tag{90}$$

$$\leq \frac{1}{2\beta_0} |v_m|_2 |\mathrm{Diag}(v_m)^2 W_m v_m|. \tag{91}$$

Notice that the second inequality holds as a direct result for Lemma A.8.

Then,

$$|\mathrm{Diag}(v_m)^2 W_m v_m|_2 = \sqrt{\sum_{i=1}^m (\sum_{j=1}^m v_{m,j}^2 W_{m,ji} v_{m,i})^2} \tag{92}$$

$$\leq \sqrt{\sum_{i=1}^m (\sum_{j=1}^m v_{m,j}^2 |W_{m,ji}| |v_{m,i}|)^2} \tag{93}$$

$$\leq \sqrt{\sum_{i=1}^m (\sum_{j=1}^m v_{m,j}^2 M_0 |v_{m,i}|)^2} \tag{94}$$

$$= M_0 \sqrt{\sum_{i=1}^m (\sum_{j=1}^m v_{m,j}^2 |v_{m,i}|)^2} \tag{95}$$

$$= M_0 \sqrt{(\sum_{j=1}^m v_{m,j}^2)^2 (\sum_{i=1}^m |v_{m,i}|^2)} \tag{96}$$

$$= M_0 (\sum_{j=1}^m v_{m,j}^2) \sqrt{(\sum_{i=1}^m |v_{m,i}|^2)} \tag{97}$$

$$\leq M_0 |v_m|_2^3 \tag{98}$$

$$\leq M_0 \Upsilon^2 |v_m|_2. \tag{99}$$

For any $0 < \Upsilon \le \sqrt{\frac{\beta_0 L}{M_0}}$, we have

$$|v_m^\top (W_m^\top \text{Diag}(v_m)^2 P_m + P_m \text{Diag}(v_m)^2 W_m) v_m| \le 2 * \frac{1}{2\beta_0} |v_m|_2 |\text{Diag}(v_m)^2 W_m v_m| \tag{100}$$

$$\le \frac{1}{\beta_0} M_0 \Upsilon^2 |v_m|_2^2 \tag{101}$$

$$\le L |v_m|_2^2. \tag{102}$$

$\square$

**Lemma A.10.** *Consider a continuous function $f : [0, \infty) \to \mathbb{R}$, assume it can be approximated by a sequence of continuous functions $\{f_m\}_{m=1}^\infty$ universally:*

$$\lim_{m \to \infty} \sup_t |f(t) - f_m(t)| = 0. \tag{103}$$

*Assume the approximators $f_m$ are uniformly exponentially decaying with the same $\beta_0 > 0$:*

$$\lim_{t \to \infty} \sup_{m \in \mathbb{N}_+} e^{\beta_0 t} |f_m(t)| \to 0. \tag{104}$$

*Then the function $f$ is also decaying exponentially:*

$$\lim_{t \to \infty} e^{\beta t} |f(t)| \to 0, \quad \forall 0 < \beta < \beta_0. \tag{105}$$

*Proof.* Given a function $f \in C([0, \infty))$, we consider the transformation $\mathcal{T}f : [0, 1] \to \mathbb{R}$ defined as:

$$(\mathcal{T}f)(s) = \begin{cases} 0, & s = 0 \\ \frac{f(-\frac{\log s}{\beta_0})}{s}, & s \in (0, 1]. \end{cases} \tag{106}$$

Under the change of variables $s = e^{-\beta_0 t}$, we have:

$$f(t) = e^{-\beta_0 t} (\mathcal{T}f)(e^{-\beta_0 t}), \quad t \ge 0. \tag{107}$$

According to uniformly exponentially decaying assumptions on $f_m$:

$$\lim_{s \to 0^+} (\mathcal{T}f_m)(s) = \lim_{t \to \infty} \frac{f_m(t)}{e^{-\beta_0 t}} = \lim_{t \to \infty} e^{\beta_0 t} f_m(t) = 0, \tag{108}$$

which implies $\mathcal{T}f_m \in C([0, 1])$.

For any $\beta < \beta_0$, let $\delta = \beta_0 - \beta > 0$. Next we have the following estimate

$$\sup_{s \in [0,1]} |(\mathcal{T}f_{m_1})(s) - (\mathcal{T}f_{m_2})(s)| \tag{109}$$

$$= \sup_{t \ge 0} \left| \frac{f_{m_1}(t)}{e^{-\beta t}} - \frac{f_{m_2}(t)}{e^{-\beta t}} \right| \tag{110}$$

$$\le \max \left\{ \sup_{0 \le t \le T_0} \left| \frac{f_{m_1}(t)}{e^{-\beta t}} - \frac{f_{m_2}(t)}{e^{-\beta t}} \right|, C_0 e^{-\delta T_0} \right\} \tag{111}$$

$$\le \max \left\{ e^{\beta T_0} \sup_{0 \le t \le T_0} |f_{m_1}(t) - f_{m_2}(t)|, C_0 e^{-\delta T_0} \right\} \tag{112}$$

where $C_0$ is a constant uniform in $m$.

For any $\epsilon > 0$, take $T_0 = -\frac{\ln(\frac{\epsilon}{C_0})}{\delta}$, we have $C_0 e^{-\delta T_0} \le \epsilon$. For sufficiently large $M$ which depends on $\epsilon$ and $T_0$, by universal approximation (Equation (103)), we have $\forall m_1, m_2 \ge M$,

$$\sup_{0 \le t \le T_0} |f_{m_1}(t) - f_{m_2}(t)| \le e^{-\beta T_0} \epsilon, \tag{113}$$

$$e^{\beta T_0} \sup_{0 \le t \le T_0} |f_{m_1}(t) - f_{m_2}(t)| \le \epsilon. \tag{114}$$

Therefore, $\{f_m\}$ is a Cauchy sequence in $C([0, \infty))$.

Since $\{f_m\}$ is a Cauchy sequence in $C([0,\infty))$ equipped with the sup-norm, using the above estimate we can have $\{\mathcal{T}f_m\}$ is a Cauchy sequence in $C([0,1])$ equipped with the sup-norm. By the completeness of $C([0,1])$, there exists $f^* \in C([0,1])$ with $f^*(0) = 0$ such that

$$\lim_{m\to\infty} \sup_{s\in[0,1]} |(\mathcal{T}f_m)(s) - f^*(s)| = 0. \tag{115}$$

Given any $s > 0$, we have

$$f^*(s) = \lim_{m\to\infty} (\mathcal{T}f_m)(s) = (\mathcal{T}f)(s), \tag{116}$$

hence

$$\lim_{t\to\infty} e^{\beta t} f(t) = \lim_{s\to 0^+} (\mathcal{T}f)(s) = f^*(0) = 0. \tag{117}$$

□

## B  MEMORY QUERY IN SENTIMENT ANALYSIS BASED ON IMDB MOVIE REVIEWS

In the following example, we show the memory function of sentiment score's for single repeated word using LSTM and fine-tuned BERT model. In Figure 5, it can be seen that the memory function of simple words such as "good" and "bad" is decaying exponentially. However, the memory of sentiment in "ha" can be complicated as it's not decaying fast. This phenomenon also holds for stacked Bidirectional LSTM models.

As a comparison, we can see the memory pattern of BERT over repeated characters or words can be decaying relatively slower. At the same time, the fluctuation of memory of "ha", "bad" and "good" can fluctuate a lot (compared with LSTM models). These phenomena indicate that the transformer-type architectures might not have exponential decaying memory issues. (See Figure 6)

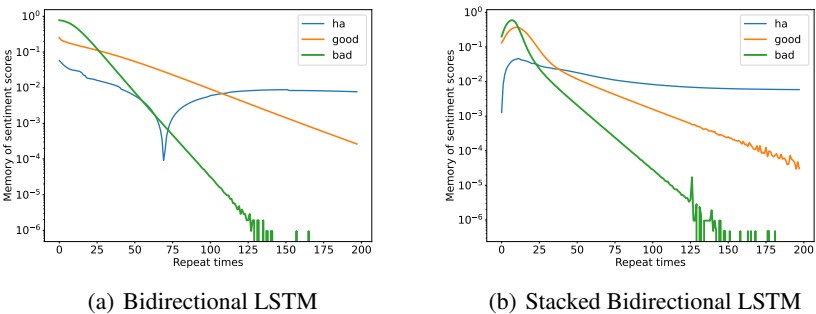

(a) Bidirectional LSTM          (b) Stacked Bidirectional LSTM

Figure 5: Memory function of sentiment scores for different words based on IMDB movie reviews using Bidirectional LSTM and stacked Bidirectional LSTM

## C  RESCALING IS NOT AN A PRIORI SOLUTION TO STABLE APPROXIMATION

Now we consider the special case: Assume $W$ is diagonal $W = diag([w_1, w_2, \dots, w_m])$. Element of matrix $U$ is 1. And the vector $c = [c_1, c_2, \dots, c_m]$.

In this section, we compare the vanilla linear RNN with a memory function $\hat{\rho}(t) = \sum_{k=1}^{m} c_k e^{w_k t}$ and a rescaled linear RNN with a memory function $\hat{\rho}(t) = \sum_{k=1}^{m} c_k e^{\frac{1}{k} w_k t}$. They are represented in the simplest form with $c_k$ and $w_k$ as the trainable weights. It can be seen that the target linear functional with memory function $\rho(t) = \sum_{k=1}^{\infty} \frac{1}{k^2} e^{-\frac{1}{k} t}$ cannot be stably approximated by linear RNN but can be stably approximated by rescaled linear RNN. (For rescaled linear RNN, one just need to pick $w_k = 1$.)

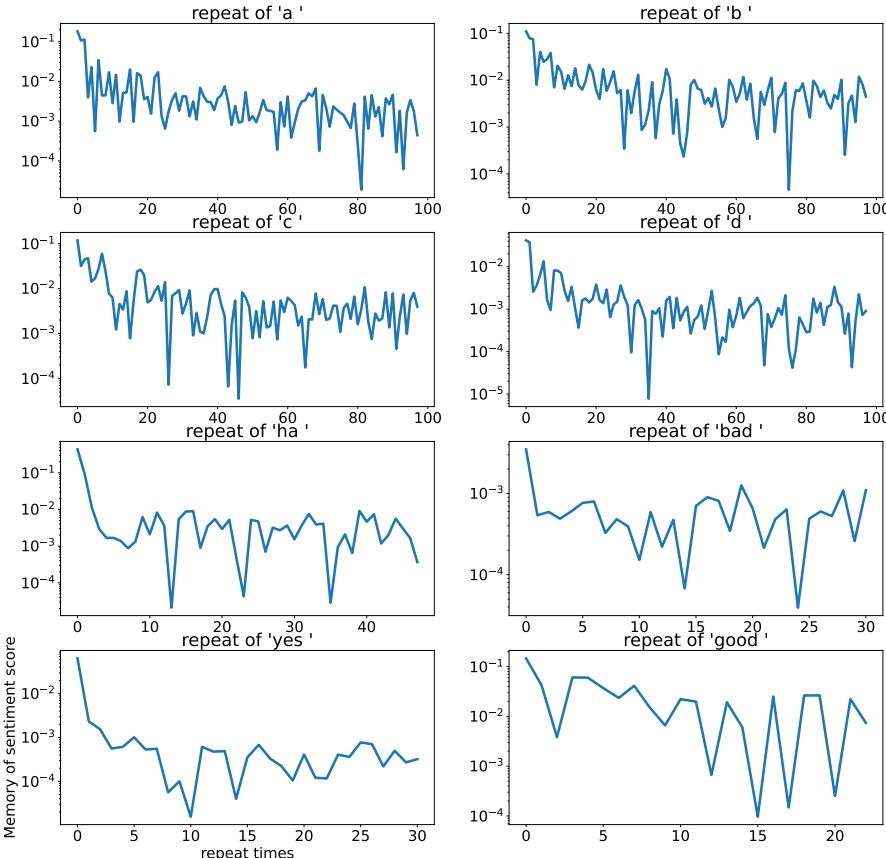

Figure 6: Memory for sentiment scores based on IMDB movie reviews
The $x$-axis is the inputs repeated times, which can be viewed as the input length. The $y$-axis is the derivative of sentiment score over repeated inputs, which is the memory of this sentiment analysis model. This tiny example implies that the memory function required in application can be in various patterns. The decaying pattern can be fluctuating in terms of the scale.

However, this rescaling does not work for all memory functions. Take the memory function $\rho(t) = \sum_{k=1}^{\infty} \frac{1}{k^2} e^{-\frac{1}{k^2}t}$ as an example, it may not be possible to stably approximate target with this memory function with linear RNN or the rescaled version.

Based on the above argument, there does not exists a rescaled RNN that can work for linear functional with arbitrary memory function. In particular, it's always feasible to construct a slow decay target that a fixed rescaled linear RNN cannot stably approximate. This result indicates that rescaling is not an a priori solution to achieve stable approximation. In practice, it's hard to get the memory function decay speed therefore it's impractical to use the rescaling method.

## D  NUMERICAL EXPERIMENTS DETAILS

Here we give the numerical experiments details for both the linear and nonlinear RNN experiments.

**Linear functionals approximated by linear RNNs**  The linear functional approximation is equivalent to the approximation of function $\rho(t)$ by exponential sum $c^\top e^{Wt} U$. For simplicity, we conduct the approximation in the interpolation approach and consider the one-dimensional input and one-dimensional output case. Furthermore we reduce the approximation problem (numerically) into the least square fitting over the discrete time grid. We fit the coefficients of polynomial and evaluate the approximation error over $[1, 2, \dots, 100]$.

The exponential decaying memory function is manually selected to show the significance across different hidden dimensions:

$$\rho(t) = 0.9^t \tag{118}$$

while the polynomial decaying memory function is $\rho(t) = \frac{1}{(t+1)^{1.1}}$. The additional 1 is added as we require the memory function to be integrable.

The perturbation list $\beta \in [0, 5.0 * 10^{-4}, 5.0 * 10^{-4} * 2^1, \ldots, 5.0 * 10^{-4} * 2^{20}]$. Each evaluation of the perturbed error is sampled with 10 different weight perturbations to reduce the variance.

The approximation of linear functional by linear RNN has simple structures. So we did not use gradient-based optimisation. The approximation of linear functional via linear RNN is equivalent to the minimisation problem $\min_{c,W,U} ||H_t - \hat{H}_t||$, which can be simplified into the following function approximation problem $\min_{c,W,U} \sup \int_0^\infty |\rho(s) - c^\top e^{sW} U| ds$. After eigenvalue decompositions of $W$ and change of variables, the above problem can be reduced to the polynomial approximation of function $\rho(-\ln t)$ over interval $(0, 1]$. Therefore we solve this approximation problem via linear regression instead of the gradient-based optimisation. The code are attached in the supplementary materials.

**Nonlinear functionals approximated by nonlinear RNNs**   Still, we consider the one-dimensional input and one-dimensional output case.   We train the model over timestamp $[0.1, 0.2, \ldots, 3.2]$ and evaluate the approximation error over the same horizon $[0.1, 0.2, \ldots, 10.0]$.

In the experiments to approximate the nonlinear functionals by nonlinear RNNs, we train each model for 1000 epochs, the stopping criterion is the validation loss achieving $10^{-8}$. The optimizer used is Adam with initial learning rate 0.005. The loss function is mean squared error. The batch size is 128 while the train set size and test set size are 12800.

The polynomial decaying memory function is $\rho(t) = \frac{1}{(t+1)^{1.5}}$ equals to 1 at $t = 0$. The only difference is the decaying speed.

The perturbation list $\beta \in [0, 10^{-11} * 2^0, 10^{-11} * 2^1, \ldots, 10^{-11} * 2^{35}]$. Each evaluation of the perturbed error is sampled with 3 different weight perturbations (and take the maximum) to reduce the variance of the perturbation error.

# E   ADDITIONAL NUMERICAL RESULTS

**Curse of memory over ReLU activation**   Despite the activation $relu$ is not smooth at $\sigma(0) = 0$, we numerically verify that the curse of memory phenomenon also holds. In Figure 7, the left RNN using ReLU cannot stably approximate target with polynomial memory while the RNN with Softplus reparameterization can stably approximate the target.

**Stable re-parameterization as a potential solution**   In Figure 4, we show that stable reparameterization such as exponential reparameterization and softplus reparameterization enables the stable approximation of linear functional with polynomial memory by linear RNNs. As shown in Figure 8, we further verify that stable re-parameterization enables the stable approximation of targets with polynomial decaying memory by hardtanh/tanh RNNs.

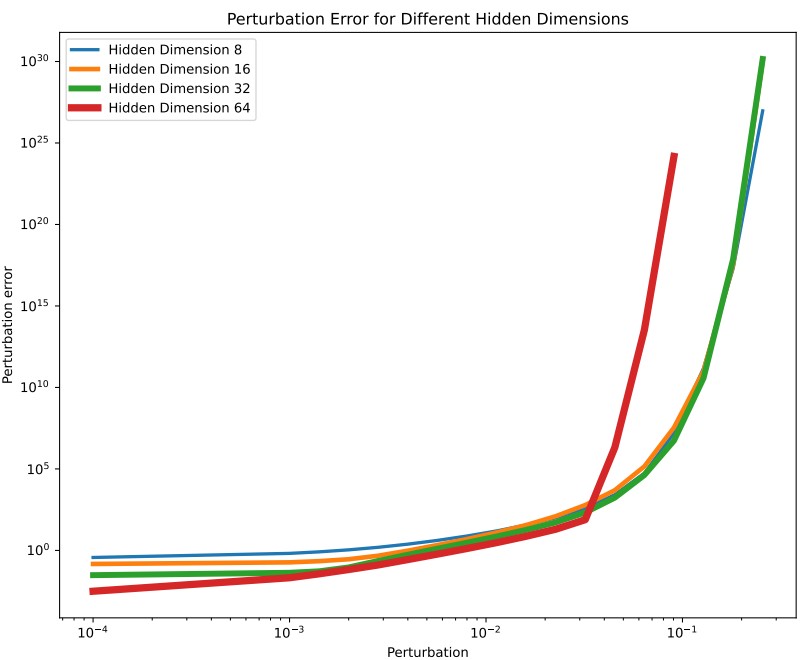

(a) Without reparameterization + ReLU RNN

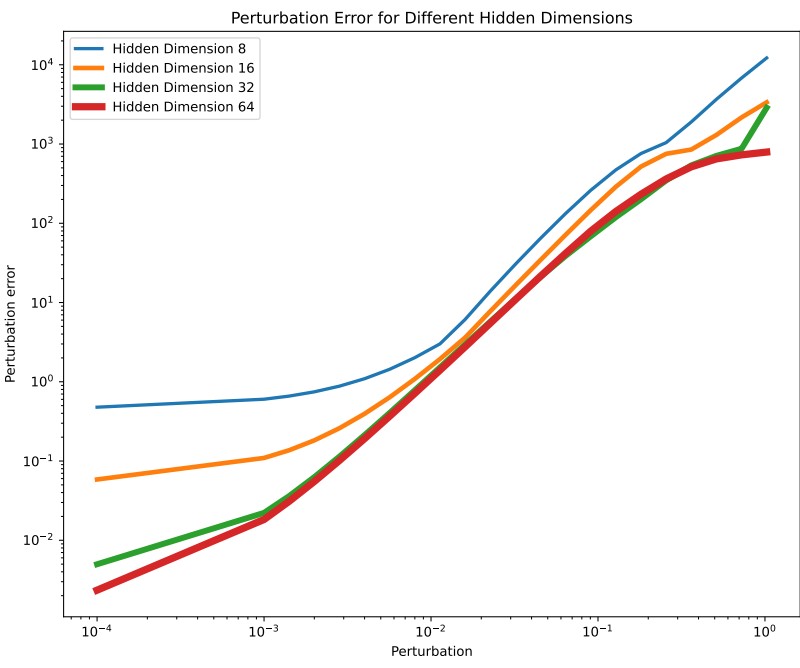

(b) Softplus reparameterization + ReLU RNN

Figure 7: When the activation used is ReLU, vanilla nonlinear RNN does not have stable approximation while the softplusRNN can be stably approximated.

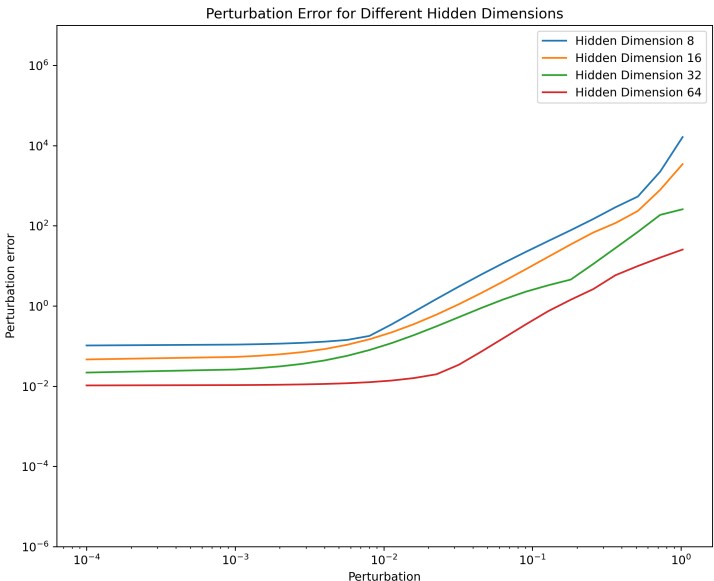

(a) Softplus reparameterization + Hardtanh RNN

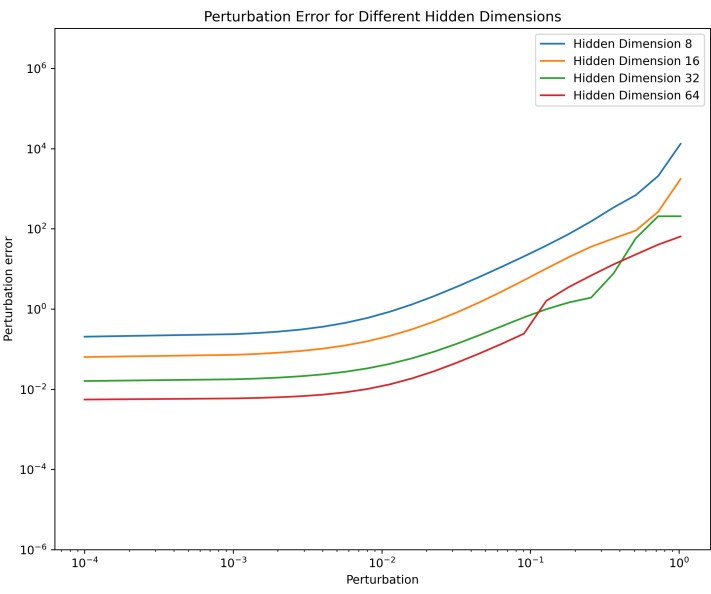

(b) Softplus reparameterization + Tanh RNN

Figure 8: Linear functionals with polynomial memory can be stably approximated by hardtanh/tanh RNNs using Softplus parameterisation for $W$. These empirical findings underscore that stable parameterisation is pivotal for attaining a stable approximation for targets with long-term memory. The experiments build upon and extend the results presented in Figure 4.

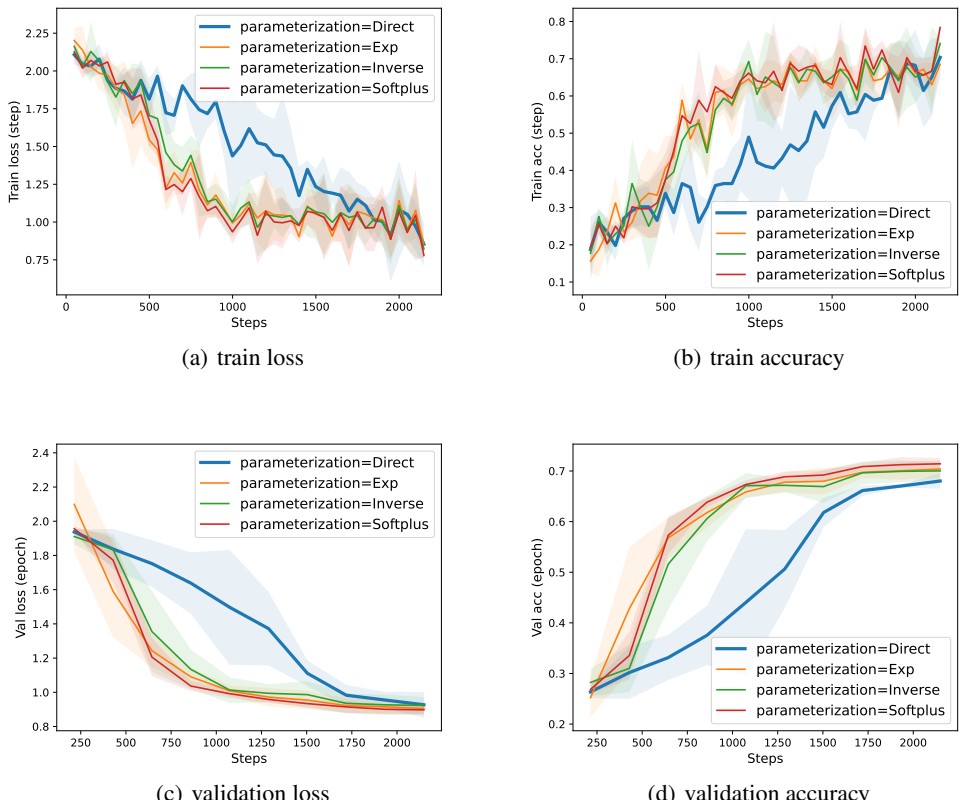

(a) train loss

(b) train accuracy

(c) validation loss

(d) validation accuracy

Figure 9: Image classification on MNIST. The performance of stable reparameterization is better than the direct parameterization of recurrent weight $g(M) = M$. Under the same weight initialization, the stable reparameterization can speed up the optimization. The shadow corresponds to the standard error evaluated over 3 repeats.

|  | Test loss(std err) | Test accuracy (std err) |
|---|---|---|
| softplus (stable) | 0.8903 (0.0004) | 71.36(0.03) |
| exp (stable) | 0.9028 (7.2E-05) | 70.55(0.01) |
| inverse (stable) | 0.9082 (0.0005) | 70.77(0.05) |
| direct (unstable) | 0.9177 (0.0105) | 68.47(0.06) |

Table 1: Test loss and accuracy of different parameterization methods over MNIST

**Stable re-parameterization facilitate the optimization**   In this subsection, we demonstrate the optimization advantages provided by stable re-parameterization in enhancing long-term memory learning. We train the nonlinear RNNs on the MNIST dataset for 10 epochs. The models are initialized with identical weight configurations, underscoring the impact of recurrent weight parameterization. As illustrated in Figure 9, the implementation of stable reparameterization significantly expedites the optimization process. It's important to note that this parameterization technique does not alter the inherent capacity of the model, resulting in comparable final performance outcomes (Table 1).

## F   MEMORY FUNCTION DEFINED VIA IMPULSE INPUTS

Besides the Heaviside inputs, it's also feasible to define the memory function via the impulse inputs. Let $x \in \mathbb{R}^d$ and consider the following impulse inputs input sequence $\mathbf{i}_t^x = \begin{cases} x & t = 0, \\ 0 & t \neq 0. \end{cases}$ Hereafter

we call the following memory function to be impulse-inputs-induced memory function.

$$\mathcal{M}(\mathbf{H})(t) := \sup_{x \neq 0} \frac{1}{|x|_\infty} \left| H_t(\mathbf{i}^x) \right|. \tag{119}$$

In the linear functional case, notice that according to Equation (3)

$$\sup_{x \neq 0} \frac{|H_t(\mathbf{i}^x)|}{\|\mathbf{i}^x\|_\infty} = \sup_{x \neq 0} \frac{|x^\top \rho(t)|}{|x|_\infty} = |\rho(t)|_1. \tag{120}$$

Based on the impulse-inputs-induced memory function, if we assume this memory function and its derivative are decaying, then we have a similar result as Theorem 3.9.

**Sketch proof for Theorem 3.9 based on impulse-inputs-induced memory function**  Assume $\beta_0$ is the stability radius, apply the proof of Theorem 3.9, we have for any $\beta < \beta_0$

$$\lim_{t \to \infty} e^{\beta t} \left| \frac{d}{dt} y_t \right| \to 0. \tag{121}$$

Fix $C > 0$, there exists $t_0$ such that for any $t \geq t_0$, $|\frac{d}{dt} y_t| \leq C e^{-\beta t}$. Since the target has decaying memory $y_\infty := \lim_{t \to \infty} y_t = 0$. Then we have

$$\lim_{t \to \infty} e^{\beta t} |y_t - y_\infty| = \lim_{t \to \infty} \left| \int_t^\infty \frac{d}{ds} y_s ds \right| \tag{122}$$

$$\leq \lim_{t \to \infty} \left| \int_t^\infty C e^{-\beta(s-t)} ds \right| \tag{123}$$

$$\leq C < \infty. \tag{124}$$

Therefore the memory function based on the impulse inputs is also decaying exponentially.

$$\lim_{t \to \infty} e^{\beta t} |y_t| \leq C. \tag{125}$$

## G  Memory function defined via reversed inputs

Let $x \in \mathbb{R}^d$ and consider the following reversed inputs input sequence $\mathbf{r}_t^x = \begin{cases} x & t < 0, \\ 0 & t \geq 0. \end{cases}$ Hereafter we call the following memory function to be impulse-inputs-induced memory function.

$$\mathcal{M}(\mathbf{H})(t) := \sup_{x \neq 0} \frac{1}{|x|_\infty} \left| \frac{d}{dt} H_t(\mathbf{r}^x) \right|. \tag{126}$$

In the linear case, notice that according to Equation (3)

$$\sup_{x \neq 0} \frac{\left| \frac{d}{dt} H_t(\mathbf{r}^x) \right|}{\|\mathbf{r}^x\|_\infty} = \sup_{x \neq 0} \frac{|x^\top \rho(t)|}{|x|_\infty} = |\rho(t)|_1. \tag{127}$$

Based on the reversed-inputs-induced memory function, if we assume this memory function and its derivative are decaying, then we have Theorem 3.9.

**Sketch proof for Theorem 3.9 based on reversed-inputs-induced memory function**  Assume $\beta_0$ is the stability radius, apply the proof of Theorem 3.9, we have for any $\beta < \beta_0$

$$\lim_{t \to \infty} e^{\beta t} \left| \frac{d}{dt} y_t \right| \to 0. \tag{128}$$

Therefore the memory function based on the reversed inputs is also decaying exponentially.
The memory function defined over reversed inputs can be **harder to evaluate** as the outputs of different input sequences $(x, 0, \dots,), (x, x, 0, \dots, 0), (x, x, x, 0, \dots, 0), \dots$ need to be evaluated.

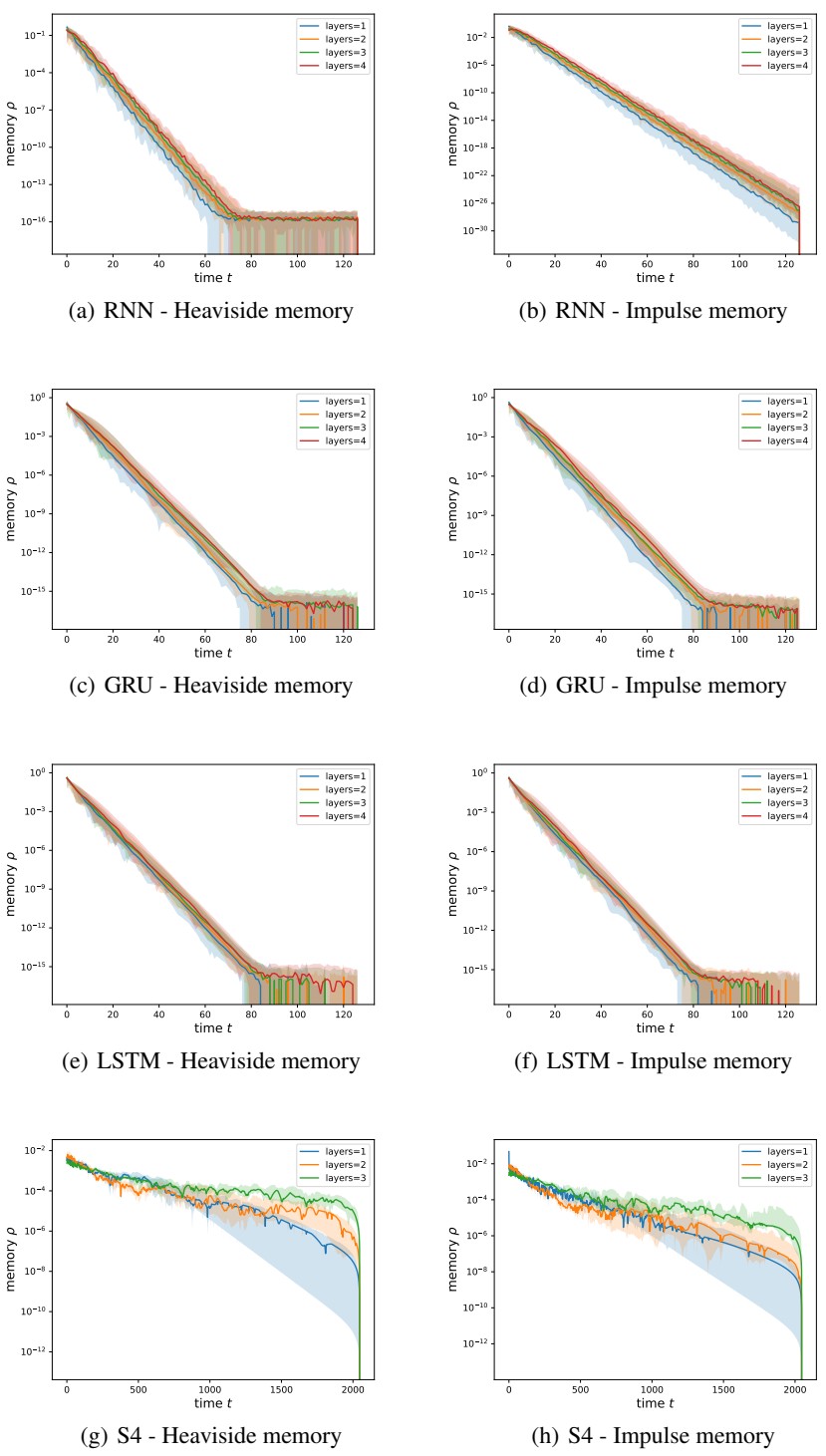

Figure 10: Memory functions of randomly-initialized recurrent networks evaluated over Heaviside inputs and impulse inputs. The memory functions evaluated over different inputs are **numerically equivalent**. The shadow indicates the error bar for 20 repeats.

## H QUALITATIVE EQUIVALENCE OF DIFFERENT MEMORY FUNCTION DEFINITIONS

Apart from the previous sections that study the equivalence of memory functions over linear functionals. In Figure 10, we show that the memory functions evaluated over different test inputs are

qualitatively equivalent. Since the reversed inputs are harder to evaluate numerically, we only compare the memory functions evaluated over Heaviside inputs and impulse inputs.

Besides the qualitative comparison that different memory functions are "equivalent", here we show they all characterize the decay speed of the model's memory of a impulse change at time $t = 0$.

**Relation between memory decaying speed and target dependence on early inputs**    An intuitive explanation is that current memory functions characterize how fast the memory about the sudden input change at $t = 0$ decays as $t$ grows. Inputs such as Heaviside inputs, reversed inputs, and impulse inputs all have this feature. Therefore the memory function evaluated over these inputs can be interpreted as **a measure for "dependence of the target function on early inputs".**

$$\rho_T := \sup_{x \neq 0} \frac{|y_T(\mathbf{i}^x)|}{|x|} = \sup_{x \neq 0} \frac{H_T((x, 0, ...))}{|x|}. \quad \text{Impulse-inputs-based memory}$$

We show the "equivalence" of different memory functions in linear functional sense in Appendix F and G. Since memory functions over Heaviside inputs are easier to evaluate numerically (as shown in Appendix H), we adopt the corresponding derivative form (as shown in Equation 5) to define the memory function.

Here we explain the reason that the memory function is evaluated over certain subclass of inputs rather than all continuous inputs:

**Why special inputs**    A natural question is why we need a particular set of inputs rather than use any continuous input. The identity functional $H_t(\mathbf{x}) = x_t$ takes the input sequence $x_t = \frac{1}{t}, t > 0$ to a polynomial decaying output $y_t = \frac{1}{t}$. However, this does not mean the functional has a polynomial-decaying memory. In fact, this identity functional has no memory of past inputs in the sense that changing $x_0$ does not influence the output of $y_t$ with $t > 0$. The above example indicates the selection of inputs to extract the memory pattern needs to be chosen properly. This is also one of the reasons we select the Heaviside inputs as the test inputs to construct the derivative form.

# I    ADDITIONAL NUMERICAL RESULTS FOR THE FILTERING EXPERIMENT

In this section, we first show the existence of nonlinewar RNNs with non-decaying memory. Therefore in Figure 3 we need to filter the targets. Then the example of learning transformer with nonlinear RNN is included. This example shows that transformer in general does not have a decaying memory. Therefore the learning of transformer can highly difficult that we cannot achieve the approximation with 1000 epochs training.

## I.1    EXISTENCE OF NONLINEAR RNNS WITHOUT DECAYING MEMORY

In Figure 3, we show that the filtering of nonlinear RNNs with stable approximation presents the target with only exponentially decaying memory. Here we present in Figure 11 that nonlinear RNNs with randomly initialized weights have diverse memory patterns. Moreover, consider the following examples.

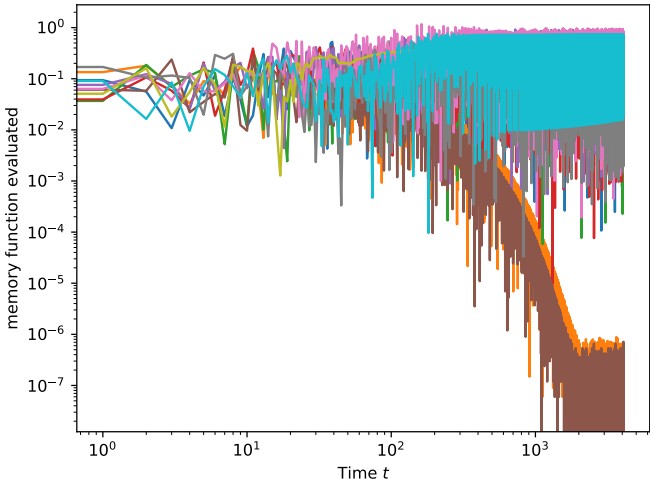

(a) Examples of nonlinear RNNs have non-decaying memory

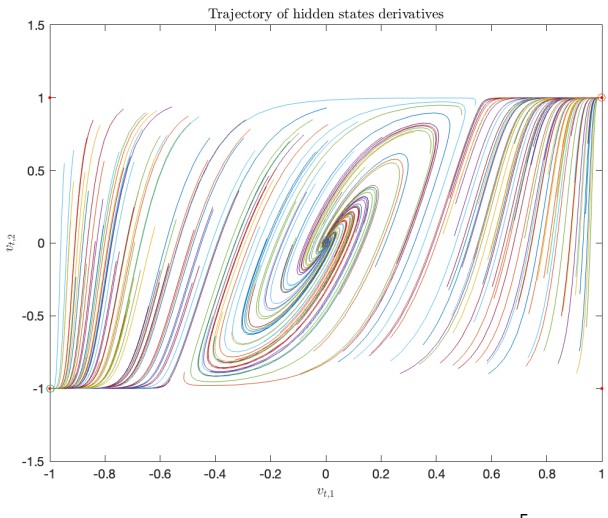

(b) Example of 2D RNN with periodic memory $W = \begin{bmatrix} 1 & 1 \\ -4 & -3 \end{bmatrix}$.

Figure 11: In the left panel, we show the existence of tanh RNNs with non-decaying memory. In the right panel, we use a specific 2D tanh RNN to show the existence of a periodic trajectory. In the figure, $[0, 0], [1, 1], [-1, -1]$ are the attractor for the trajectories of $(v_1, v_2) = (\frac{d\bar{h}_{t,1}}{dt}, \frac{d\bar{h}_{t,2}}{dt})$. Therefore the boundary of the different converging set will be a periodic trajectory over the 2D plane. **The periodic trajectory of hidden state derivative indicates a periodic memory function**.

## I.2 FILTERING TOWARDS TRANSFORMER

Apart from the nonlinear RNNs targets, we also construct targets with polynomial decaying memory. It can be seen in Figure 12 that the transformers with non-decaying memory function cannot be stably approximated by tanh RNNs.

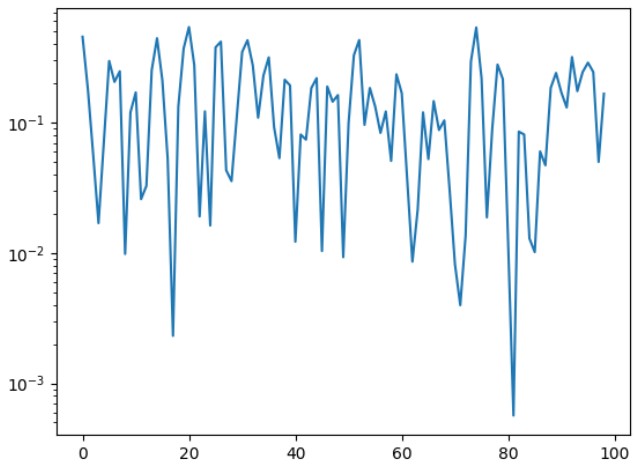

(a) Memory function of the target randomly initialized transformer

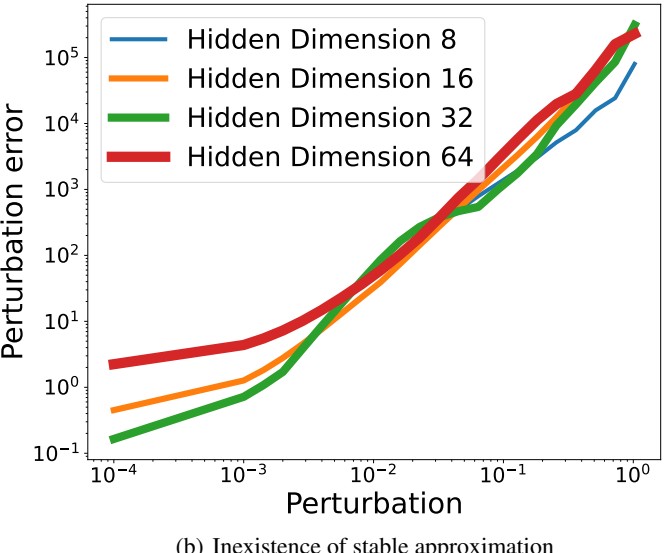

(b) Inexistence of stable approximation

Figure 12: If the target functional does not have an exponentially decaying memory, then the targets cannot be stably approximated. The largest RNN with hidden dimension $m = 64$ failed to achieve an approximation error smaller than the case for $m = 32$ and $m = 16$.

## J    DISCUSSION ON EXTENDING THE THEOREM 3.9 TO GRU/LSTM

We will analyze GRU as LSTM comes with more complicated structure. The classical GRU is of the following structure:

$$z_t = \sigma(W_z x_t + U_z h_{t-1} + b_z), \tag{129}$$

$$r_t = \sigma(W_r x_t + U_r h_{t-1} + b_r), \tag{130}$$

$$\hat{h}_t = \phi(W_h x_t + U_h(r_t \odot h_{t-1}) + b_h), \tag{131}$$

$$h_t = (1 - z_t) \odot h_{t-1} + z_t \odot \hat{h}_t. \tag{132}$$

Here $\sigma$ is sigmoid activation and $\phi$ is the tanh activation. The hidden state equations can be reformulated into

$$h_t = h_{t-1} + z_t \odot (\hat{h}_t - h_{t-1}). \tag{133}$$

The corresponding continuous-time version can be given by

$$\frac{dh_t}{dt} = z_t \odot (\hat{h}_t - h_t) \tag{134}$$

$$= \sigma(W_z x_t + U_z h_t + b_z) \odot (\hat{h}_t - h_t) \tag{135}$$

$$= \sigma(W_z x_t + U_z h_t + b_z) \odot (\phi(W_h x_t + U_h(r_t \odot h_t) + b_h) - h_t) \tag{136}$$

$$= \sigma(W_z x_t + U_z h_t + b_z) \tag{137}$$

$$\odot (\phi(W_h x_t + U_h(\sigma(W_r x_t + U_r h_t + b_r) \odot h_t) + b_h) - h_t). \tag{138}$$

Let $x \in \mathbb{R}^d$ and we consider the Heaviside inputs $\mathbf{u}_t^x = x \cdot \mathbf{1}_{[0,\infty)}(t) = \begin{cases} x & t \geq 0, \\ 0 & t < 0. \end{cases}$ After change of variable over the bias term $b_z := W_z x + b_z, b_r := W_r x + b_r, b_h := W_z x + b_h$, the corresponding hidden dynamics can be simplified into

$$\frac{dh_t}{dt} = \sigma(U_z h_t + b_z) \tag{139}$$

$$\odot (\phi(U_h(\sigma(U_r h_t + b_r) \odot h_t) + b_h) - h_t), \tag{140}$$

Notice that $h^* = 0$ is an equilibrium point if $b_h = 0$:

$$\sigma(b_z) \odot (\phi(U_h(\sigma(b_r) \odot 0) + 0) - 0) = 0. \tag{141}$$

Similar linearization can be utilized to analyse the corresponding first-order system:

$$\frac{dh_t}{dt} = \text{Diag}(\sigma(b_z))(U_h \text{Diag}(\sigma(b_r)) - I)h_t. \tag{142}$$

Our stable approximation argument can be used to show $\text{Diag}(\sigma(b_z))(U_h \text{Diag}(\sigma(b_r)) - I)$ is Hurwitz. However, $h^* = 0$ might not be the only equilibrium point. Furthermore, when $b_h \neq 0$, it's harder to find all equilibrium points therefore further analysis are needed.

