# OpenReview forum: "Inverse Approximation Theory for Nonlinear Recurrent Neural Networks"
_ICLR.cc/2024/Conference — ICLR 2024 spotlight_

### Official Review · Reviewer_6cRG · 2023-10-22

**Soundness:** 3 good
**Presentation:** 2 fair
**Contribution:** 3 good
**Rating:** 8
**Confidence:** 3

**Summary:**

This paper investigates the inverse approximation theory for nonlinear recurrent neural networks (RNN) and extends the curse of memory from linear RNNs to nonlinear RNNs. The contribution of this paper can be summarized as follows.
1. Authors define the memory of nonlinear RNNs, which is consistent with the memory of linear RNNs.
2. Authors introduce a notion of stable approximation.
3. Authors prove a Bernstein-type approximation theorem for nonlinear functional sequences through nonlinear RNNs, which says that if a target function can be stably approximated by nonlinear RNNs, then the target function must have exponential decaying memory.
4. Based on the theoretical result, the authors propose a suitable parameterization that enables RNN to stably approximate target functions with non-exponential decaying memory.

**Strengths:**

1. To the best of my knowledge, this is the first paper studying the inverse approximation theorem for nonlinear RNNs.
2. The approximation of neural networks determines the existence of a low-error solution and has been widely investigated. The inverse approximation theorem, which is harder and less studied, concerns the efficiency of approximation, which is a crucial aspect of utilizing neural networks in practice.
3. The nonlinearity is of central interest in empirical implementations and requires more challenging proofs.

**Weaknesses:**

1. The definition of memory for nonlinear RNNs is consistent with that for linear RNNs, but the rationality of the proposed definition needs more explanations. The authors review the definition of memory for linear RNNs and observe that if the memory decays fast, then the target has short memory, and vice versa. The observation can be obtained directly from the definition and satisfies the intuition of memory. But for the memory for nonlinear RNNs, authors motivate it from a derivative form. This motivation is too mathematical and lacks an intuitive explanation. What is the relationship between the decaying speed of memory and the dependence of the target function on early inputs?
2. The parameterization motivated by the theories should be explained in more detail. The parameterization is based on an important claim that the eigenvalue real part being negative leads to stability, which is not explained in section 3. This makes the logic in section 3.4 incomplete and the parameterization hard to understand. Can authors provide an intuitive explanation for this?
----
The authors' answers are convincing and I have updated my ratings.

**Questions:**

See weakness.

---

> ### Author Response · Authors · 2023-11-18
>
> We thank the reviewer for their thoughtful comments and constructive feedback.
>
> Weaknesses:
>
> 1. `The definition of memory for nonlinear RNNs is consistent with that for linear RNNs, but the rationality of the proposed definition needs more explanations.`
>
> (**Rationality of the proposed definition**)
> People use the outcomes of long-term memory tasks to demonstrate the long-term memory of models.
> Although the performance in common experiments, such as addition problems, copying tasks, and associative recalls is indicative of long-term memory attributes, it is important to note that these experiments are heuristic in nature.
> We propose the memory function definitions for a precise, task-independent characterization of the model memories. (Definition 3.1 Page 5)
>
> `The authors review the definition of memory for linear RNNs and observe that if the memory decays fast, then the target has short memory, and vice versa. The observation can be obtained directly from the definition and satisfies the intuition of memory. But for the memory for nonlinear RNNs, authors motivate it from a derivative form. This motivation is too mathematical and lacks an intuitive explanation. What is the relationship between the decaying speed of memory and the dependence of the target function on early inputs?`
>
> (**Relation between memory decaying speed and target dependence on early inputs**)
> An intuitive explanation is that current memory functions characterize how fast the memory about the sudden input change at $t=0$ decays as $t$ grows.
> Inputs such as Heaviside inputs, reversed inputs, and impulse inputs all have this feature.
> Therefore the memory function evaluated over these inputs can be interpreted as **a measure for "dependence of the target function on early inputs".**
> $$\rho_T := \sup_{x \neq 0} \frac{|y_T(\mathbf{i}^x)|}{|x|}  = \sup_{x \neq 0} \frac{H_T((x, 0, ...))}{|x|}. \quad \textrm{Impulse-inputs-based memory}$$
> We show the "equivalence" of different memory functions in linear functional sense in Appendix F and G.
> Since memory functions over Heaviside inputs are easier to evaluate numerically (as shown in Appendix H), we adopt the corresponding derivative form (as shown in Equation 5) to define the memory function.
>
> (**Why special inputs**)
> A natural question is why we need a particular set of inputs rather than use any continuous input.
> The identity functional $H_t(\mathbf{x}) = x_t$ takes the input sequence $x_t = \frac{1}{t}, t > 0$ to a polynomial decaying output $y_t = \frac{1}{t}$. However, this does not mean the functional has a polynomial-decaying memory. In fact, this identity functional has no memory of past inputs in the sense that changing $x_0$ does not influence the output of $y_t$ with $t > 0$.
> The above example indicates the selection of inputs to extract the memory pattern needs to be chosen properly.
> This is also one of the reasons we select the Heaviside inputs as the test inputs to construct the derivative form.

---

> ### Author Response · Authors · 2023-11-18
>
> 2. `The parameterization motivated by the theories should be explained in more detail. The parameterization is based on an important claim that the eigenvalue real part being negative leads to stability, which is not explained in section 3. This makes the logic in section 3.4 incomplete and the parameterization hard to understand. Can authors provide an intuitive explanation for this?`
>
> Thank you for raising this constructive question.
>
> (**Connection between negative eigenvalue real parts and stability**)
> For continuous-time linear RNN, the bounded-input bounded-output stability is **equivalent** to eigenvalues of $W$ having negative real parts.
> Our proof of Bernstein theorem shows that the eigenvalues real part being negative is a necessary condition for stability of the continuous-time nonlinear RNN (see Lemma A.7 in Appendix A.10 Proof of Lemmas).
> Therefore the **stability boundary** for eigenvalues of $W$ is the line in complex plane with zero real parts $\{z|\textrm{Re}(z)=0\}$.
>
> (**Intuitive explanation of the effects of reparameterization**)
> Without reparameterization, the approximation of long-term memory requires the eigenvalue real parts come close to 0 from the negative direction, while the stability requires the eigenvalues bounded away from 0.
> Therefore the learning of long-term memory cannot be achieved stably.
> With stable reparameterization, the approximation of long-term memory still requries the eigenvalues to be close to 0, but the stability does not enforce the eigenvalues to be bounded away from 0.
> Therefore it's possible to learn long-term memories stably.
>
> (**Motivation for reparameterization**)
> Theorem 3.9 proves that if the eigenvalues are bounded away with distance $\beta$ from the stability boundary, then the memory decays exponentially with speed $e^{- \beta t}$.
> Therefore, when models are learning targets with long-term memories, the eigenvalues are required to come close to the stability boundary.
> In Section 3.4 (Page 7 and 8), we show that stable reparameterization allows the eigenvalues to come close to the stability boundary $\{z|\textrm{Re}(z)=0\}$ without having the stability issue.
>
> (**Empirical evidence**)
> The reparameterization method has been adopted in various previous works ([1,2,3,4,5]) to improve the optimization of recurrent models such as spectral RNN and state-space models.
> However, the previous works usually attribute the performance to the optimization stability of reparameterization.
> Our work highlights that stable reparameterization plays a dual role: it not only contributes to the stable training of dynamical system, but is also **necessary for the learning of long-term memories**.
>
> ---
>
> References:
>
> [1]. (SVD parameterization) Zhang, Jiong, Qi Lei, and Inderjit Dhillon. "Stabilizing gradients for deep neural networks via efficient svd parameterization." In International Conference on Machine Learning, pp. 5806-5814. PMLR, 2018.
>
> [2]. (S4) Albert Gu, Karan Goel, and Christopher Re. "Efficiently Modeling Long Sequences with Structured State Spaces." In International Conference on Learning Representations. 2021
>
> [3]. (DSS) Gupta, Ankit, Albert Gu, and Jonathan Berant. "Diagonal state spaces are as effective as structured state spaces." Advances in Neural Information Processing Systems 35 (2022): 22982-22994.
>
> [4]. (S5) Smith, Jimmy TH, Andrew Warrington, and Scott Linderman. "Simplified State Space Layers for Sequence Modeling." In The Eleventh International Conference on Learning Representations. 2022.
>
> [5]. (LRU) Orvieto, Antonio, Samuel L. Smith, Albert Gu, Anushan Fernando, Caglar Gulcehre, Razvan Pascanu, and Soham De. "Resurrecting recurrent neural networks for long sequences." arXiv preprint arXiv:2303.06349 (2023).

---

> > ### Comment · Reviewer_6cRG · 2023-11-20
> >
> > Thanks for your answers. I am convinced and have updated my rating.

---

### Official Review · Reviewer_dqeg · 2023-10-27

**Soundness:** 4 excellent
**Presentation:** 4 excellent
**Contribution:** 3 good
**Rating:** 8
**Confidence:** 1

**Summary:**

The paper introduces an inverse approximation theorem for non-linear recurrent neural networks, extending known, results for the linear case. After introducing new tools that enable analyzing nonlinear dynamics, it shows a somewhat negative result that nonlinear RNNs suffer from the same curse of memory as linear RNNs. Finally, the authors suggest a type of reparametrization that could possibly enable stability and good approximation properties.

**Strengths:**

As an outsider to the field, the paper introduces/uses formal concepts that I find insightful:

- The memory function that is used in the paper mathematically characterizes the intuitive behavior of RNNs.
- The notion of stable approximation is an interesting proxy for how a function can be learned by gradient descent which, from my very limited knowledge, seems to be rare in approximation theory.

Overall, I found overall the paper well written, in a way that is accessible to a decently large audience.

**Weaknesses:**

I am not qualified enough to identify the weaknesses of the paper.

**Questions:**

I am a bit confused with the Heaviside input at the end of Page 4. Intuitively, I would have put it the other way around: the network is submitted to an input $x$ until 0, and then evolves autonomously later. This would allow us to measure how fast the network forgets about $x$. Can you elaborate on the choice in the paper?

The reparametrization that you propose reminds me of the one used in the LRU architecture (Orvieto et al. 2023, ICML) and present in some deep-state space models (see references in the LRU paper), it may be nice to mention this. I’m not sure I fully understand what your theoretical results show for the reparametrization. To the best of my understanding, even with such a reparametrization, RNN would still only be able to learn in the same manner targets with decaying memory. Can you elaborate more on that?

---

> ### Author Response · Authors · 2023-11-18
>
> We thank the reviewer for the positive feedback.
>
> Questions:
>
> 1. `I am a bit confused with the Heaviside input at the end of Page 4. Intuitively, I would have put it the other way around: the network is submitted to an input $x$ until 0, and then evolves autonomously later. This would allow us to measure how fast the network forgets about $x$. Can you elaborate on the choice in the paper?`
>
> Thank you for proposing the new scheme for memory function evaluation.
> We hereafter call the new input scheme "reversed inputs".
> Let $\mathbf{x}$ be constant inputs and $\mathbf{u}^x$ be Heaviside inputs, the reversed inputs can be written as
> $$\mathbf{r}^x = \mathbf{x} - \mathbf{u}^x.$$
> The new scheme can also be used to define a memory function and establish a similar result as Theorem 3.9 (see Appendix G, Page 28).
> Here we show the equivalence to our definition in the following sense:
>
> (**Equivalence over linear functionals**)
> Based on the linear functional representation, $H_t(\mathbf{r}^x) = C - H_t(\mathbf{u}^x)$ for some constant $C$.
> Therefore the memory functions defined via $|\frac{dH_t}{dt}|$ are the same for both Heaviside inputs and reversed inputs.
> They can extract the same memory function $|\rho|$ of the linear functional.
>
> (**Numerical equivalence over nonlinear functionals**)
> In the nonlinear case, they are not exactly the same but the empirical experiments indicate the qualitative behaviours are usually the same.
> In Appendix H, Figure 10 (Page 29), we construct randomly-initialized recurrent models and evaluate the memory functions over different inputs to demonstrate the numerical equivalence.
> This similarity indicates that memory function is a good surrogate tool to compare the memory patterns.
>
> (**Intuitive explanation of equivalence**)
> For both Heaviside and reversed inputs, the input change occurs at time $t=0$.
> As time $T$ increases, the memory function $\rho(T)$ reflects the impact of input change after time $T$.
> Therefore the Heaviside inputs and reversed inputs are equivalent in characterizing the long-term memory effects.
>
> One of the reasons to choose Heaviside inputs over reversed inputs is that the Heaviside inputs are easier to evaluate numerically (see Appendix H, Page 28-30).

---

> ### Author Response · Authors · 2023-11-18
>
> 2. `The reparametrization that you propose reminds me of the one used in the LRU architecture (Orvieto et al. 2023, ICML) and present in some deep-state space models (see references in the LRU paper), it may be nice to mention this. I’m not sure I fully understand what your theoretical results show for the reparametrization. To the best of my understanding, even with such a reparametrization, RNN would still only be able to learn in the same manner targets with decaying memory. Can you elaborate more on that?`
>
> We have added the LRU [1] and related works [2,3,4,5] to the stable reparameterization discussion in Section 3.4.
>
> The reparameterization does not fundamentally change the models' exponential decay pattern.
> Both Theorem 3.9 and Section 3.4 have no contradiction with this result.
> Theorem 3.9 shows that without reparameterization, when learning long-term memory, one cannot achieve stability and approximation at the same time.
> But the inverse approximation theorem does not limit the memory pattern of reparameterized models.
> **Section 3.4 illustrates that with stable reparameterization, we can achieve the approximation of long-term memory without sacrificing the stability.**
> In particular, in Figure 4, we show that the approximation of polynomial-memory target can be achieved stably with the RNN using exponential or softplus parameterization.
>
>
>
> ---
>
> References:
>
> [1]. (LRU) Orvieto, Antonio, Samuel L. Smith, Albert Gu, Anushan Fernando, Caglar Gulcehre, Razvan Pascanu, and Soham De. "Resurrecting recurrent neural networks for long sequences." arXiv preprint arXiv:2303.06349 (2023).
>
> [2]. (HIPPO) Gu, Albert, Tri Dao, Stefano Ermon, Atri Rudra, and Christopher Ré. "Hippo: Recurrent memory with optimal polynomial projections." Advances in neural information processing systems 33 (2020): 1474-1487.
>
> [3]. (S4) Albert Gu, Karan Goel, and Christopher Re. "Efficiently Modeling Long Sequences with Structured State Spaces." In International Conference on Learning Representations. 2021
>
> [4]. (S5) Smith, Jimmy TH, Andrew Warrington, and Scott Linderman. "Simplified State Space Layers for Sequence Modeling." In The Eleventh International Conference on Learning Representations. 2022.
>
> [5]. (Universality and memory decay of SSM) Wang, Shida, and Beichen Xue. "State-space Models with Layer-wise Nonlinearity are Universal Approximators with Exponential Decaying Memory." Advances in neural information processing systems 37 (2023).

---

> > ### Comment · Reviewer_dqeg · 2023-11-20
> >
> > Thank you for your answers. The paper seems to be a solid contribution, I have increased my rating to 8.

---

### Official Review · Reviewer_aNtd · 2023-10-27

**Soundness:** 4 excellent
**Presentation:** 4 excellent
**Contribution:** 3 good
**Rating:** 8
**Confidence:** 4

**Summary:**

This paper demonstrates a Bernstein-type result for nonlinear Recurrent Neural Networks, meaning that it characterizes the type of functions that can be approximated by RNNs. The result shows that if a target sequence can be approximated by a RNN, then it has an exponentially decaying memory (for a notion of memory made precise in the paper). A similar result was proven in the literature for linear RNNs and this paper extends the result to the non-linear case.

**Strengths:**

The paper is well presented and pleasant to read. Extending previously known results to the non-linear case is interesting and brings added value. It gives a theoretical framework to understand the common saying that RNNs are unable to learn long-time dependencies in time series. A modification of the parametrization of RNNs is proposed to remedy this problem.

Technically, the paper introduces a new notion of memory that holds for nonlinear functionals and which extends the linear case, as well as a notion of stable approximation (which corresponds to the existence of a ball of parameters with uniformly good approximation properties). This proof technique is interesting and could perhaps be applied to other architectures.

Experiments support the theoretical findings.

**Weaknesses:**

I have no strong reservations about the paper. I do have a question about Figure 3, which I did not understand. I am willing to raise the rating should this interrogation be answered. [Update on Nov. 18: the authors clarified this point in their comments below, and I updated my rating accordingly].

I do not understand the filtering part of the experiment. Since we know from the theoretical results that RNNs can only represent exponentially decaying functions, the teacher models should all be exponentially decaying? So why is any filtering required? Furthermore, if indeed filtering is required, is it also performed before plotting the left-side plot? Otherwise we should be seeing some non-converging data points?
As a consequence, I am wondering if the takeaway of the experiment should be “stable approximation implies exponential decay” or rather “exponential decay implies stable approximation”.

**Minor remarks that do not influence the rating**
+ Page 5: I found the paragraph above Definition 3.2 hard to follow. I am not sure to understand the reason for introducing the terminology  “queried memory”. The experiments of Appendix B are important in my opinion because they suggest that RNNs may share the exponential decay property with LSTMs, but not Transformer. But I do not understand why they are introduced at this point in the paper, and not later on, e.g. in the conclusion.
+ Page 7: the proof sketch is difficult to follow, and I am not sure it brings significant added value. If you need additional space in the main paper, I’d suggest putting the proof sketch in the Appendix just before the proof.
+ Page 7, line -2: “In order” -> “in order”
+ Page 8: “approaching 0” -> “approaching 0 on the negative side”?
+ Page 8: “Take exponential…” -> “Taking exponential”
+ Page 8, figures 2 and 3: please avoid using plus sign and arrow inside the text. Consider replacing it by “and” and “implies”.
+ Page 16: “acivations” -> “activations”
+ Page 16: could you briefly explain or give a reference for Lyapunov equations?

**Questions:**

Do the authors think their proof technique could also be applied to LSTMs/GRUs? I imagine there would be many technical difficulties, but conceptually do the authors think it would be possible? Giving this insight in the paper might be interesting for readers.

See also weaknesses.

---

> ### Author Response · Authors · 2023-11-18
>
> We thank the reviewer for the positive feedback.
>
> Weaknesses:
>
> `I have no strong reservations about the paper. I do have a question about Figure 3, which I did not understand. I am willing to raise the rating should this interrogation be answered.`
>
> `I do not understand the filtering part of the experiment. Since we know from the theoretical results that RNNs can only represent exponentially decaying functions, the teacher models should all be exponentially decaying? So why is any filtering required? Furthermore, if indeed filtering is required, is it also performed before plotting the left-side plot? Otherwise we should be seeing some non-converging data points? As a consequence, I am wondering if the takeaway of the experiment should be “stable approximation implies exponential decay” or rather “exponential decay implies stable approximation”.`
>
> Yes, the takeaway for the filtering experiment is "stable approximation and decaying memory implies exponential decay".
> This paper primarily focuses on the inverse-type approximation theorem.
> The assertion that "Exponential decay of the target implies stable approximation" is a forward-type (Jackson-type) result that has not been established as of now.
>
> In our theory, we show that nonlinear RNNs with decaying memory are decaying exponentially.
> Randomly initialized nonlinear RNNs might not have decaying memory.
> In Appendix I.1 (Page 30) and Figure 11 (Page 31), we present examples of nonlinear RNNs having non-decaying memory.
> In particular, in Figure 11 (b), the trajectory of hidden state derivatives indicate the existence of periodic memory function from the 2D tanh RNN.
> Therefore, we construct several targets in the experiment for Figure 3 and use decaying memory and existence of stable approximation by smaller RNNs as the criterions to filter the targets teacher models.
>
> The reason we pick larger RNNs as teacher models is that nonlinear RNNs are universal approximators.
> Therefore from the density perspective, larger models with randomly initialized weights should have more diverse memory pattern than the smaller models.
> We also investigate the memory patterns of transformers.
> It can be seen in Appendix I.2 (Page 31 and 32) that transformers do not have a decaying memory so the stable approximation cannot be achieved by nonlinear RNNs: We failed to achieve a smaller validation loss with larger models ($m=64$) over 1000 epochs of training.
>
>
>
> `Minor remarks`: Thank you for the detailed writing suggestions and pointing out the typos, we have corrected them in the paper revision (marked in blue).
> 1. Page 5: The queried memory is proposed as an empirical evaluation of the target memory.
>     As our memory function is defined by a derivative form, the queried memory is the memory evaluated in practice to approximate the memory function.
>     In Appendix F and G (Page 28), we discuss the memory function queried by other test inputs and the equivalence over linear functional are discussed.
>     Numerical equivalence is demonstrated with the memory function of the randomly-initialized models in Appendix H (Page 29).
> 2. Page 7: We move the proof sketch to the Appendix and replace it with the discussion on the extension to GRU/LSTM.
> 3. Page 16: Lyapunov equations: $W_m^T P_m + P_m W_m = -Q_m$ is a commonly used equation in control theory and stability analysis.
>     We add the reference for this equation from Nonlinear System [1].

---

> ### Author Response · Authors · 2023-11-18
>
> Questions:
>
> 1. `Do the authors think their proof technique could also be applied to LSTMs/GRUs? I imagine there would be many technical difficulties, but conceptually do the authors think it would be possible? Giving this insight in the paper might be interesting for readers.`
>
> The proof of Theorem 3.9 relies on the limiting behaviour analysis around the equilibrium points.
> **We give a brief discussion for the analysis for GRU, the detailed calculation is given in Appendix J** (Page 33).
> For nonlinear RNNs, the hidden state dynamics over Heaviside inputs $\mathbf{u}^x$ is
>     $$\frac{dh_t}{dt} = \sigma(Wh_t + b).$$
> For GRU, based on the update rule $h_t = (1-z_t) \odot h_{t-1} + z_t \odot \hat{h_t} $ and corresponding definition for gates $z_t$, the continuous hidden state dynamics over Heaviside inputs is
>     $$\frac{dh_t}{dt} =\sigma(U_z h_t + b_z) \odot (\phi(U_h (\sigma(U_r h_t + b_r) \odot h_t) + b_h) - h_{t}).$$
> For nonlinear RNNs, the asymptotic behaviour can be characterized via linearization around equilibrium points.
> However, the hidden dynamics of GRU are more complex.
> If $b_h = 0$, we show $h^*=0$ is an equilibrium point.
> $$\sigma(b_z) \odot (\phi(U_h (\sigma(b_r) \odot 0) + 0) - 0) = 0.$$
> The same linearization method from Theorem 3.9 can be used and the assumption of stable approximation requires $\textrm{Diag}(\sigma(b_z)) (U_h \textrm{Diag}(\sigma(b_r)) - I)$ to be Hurwitz.
> Similar exponential decay memory can be obtained around the analysis of $h^*=0$.
> However, $h^*=0$ might not the only equilibrium points.
> The general case with $b_h \neq 0 $ is harder to analyze as the equilibrium points are much harder to solve.
> It requires more techniques to decide the convergence around equilibrium points for GRU.
> The case for LSTM is more complex as the model structure is more complicated than that of RNN and GRU.
>
> To summarize, our techniques can be applied to GRU or LSTM but further analyses are needed to extend the result.
>
> ---
> References:
>
> [1]. Hassan K. Khalil. Nonlinear systems third edition (2002), 2002

---

> > ### Comment · Reviewer_aNtd · 2023-11-18
> >
> > Thank you for your detailed answer that clarifies all my questions. I have updated my rating.

---

### Official Review · Reviewer_NURo · 2023-10-30

**Soundness:** 3 good
**Presentation:** 3 good
**Contribution:** 2 fair
**Rating:** 5
**Confidence:** 2

**Summary:**

The authors provides an inverse approximation theory for a certain class of nonlinear recurrent neural network (RNN), showing that, for the class of nonlinear RNN to approximate the sequence, the sequence should have an exponential decaying memory structure. It demonstrates the difficulties of RNN in learning and representing a longer-time dependence in the data.

**Strengths:**

The paper is relatively well written and it extends the previous theories in linear RNN. While heuristically it is not difficult to show that a popular RNN, such as LSTM or GRU, has an exponentially decaying memory, by approximating the internal state of the RNN with a relaxation equation, this manuscript provides a robust proof about such behavior.

**Weaknesses:**

While it the manuscript is clearly written with well defined problem set up, my concern is the fit to the scope of ICLR. On one hand, I believe that the analysis and definitions provided in the manuscript can be of interest in developing theories about RNN. On the other hand, the scope of the manuscript is too narrow and it is unclear what insight this study provides to benefit a broader class of RNNs or Deep Learning in general. I believe that the authors need to consider giving more insights from their results for a broader context. The stable parameterization section is not very convincing. Can you give a more concrete example about the effects of the reparameterization? Possibly, using a dynamical system or real data?

**Questions:**

NA

---

> ### Author Response · Authors · 2023-11-18
>
> We thank the reviewer for their thoughtful comments and constructive feedback.
>
> Weaknesses:
>
> 1. `While it the manuscript is clearly written with well defined problem set up, my concern is the fit to the scope of ICLR. On one hand, I believe that the analysis and definitions provided in the manuscript can be of interest in developing theories about RNN. On the other hand, the scope of the manuscript is too narrow and it is unclear what insight this study provides to benefit a broader class of RNNs or Deep Learning in general.`
>
> The scope of this paper is the theoretical analysis of nonlinear RNNs' memory pattern.
> This paper presents the first inverse approximation result for nonlinear RNNs.
> It is important for the understanding and comparison of model architectures.
> In particular, our paper points out the memory limitation of nonlinear RNNs does not only come from the optimization difficulty (such as vanishing/exploding gradient issues [1, 2]), but also more importantly comes from the model architecture.
>
> The important practical insights are as follows.
> - We define a memory function for nonlinear functionals that can be used to compare the memory patterns of different targets and models.
> - We give a definition of stable approximation, which is a more suitable approximation assumption in the setting of gradient-based optimization.
> - We prove the first inverse approximation theorem that shows the memory limitation of nonlinear RNNs comes from the model architecture.
> - Our analysis points out the stable parameterization as a solution to memory limitation of the architecture.
>
> The memory function and stable approximation concepts can be adopted in the study of **any** sequence-to-sequence relationships.
> Based on these two concepts, we prove in Theorem 3.9 (Page 7) that nonlinear functionals learned by nonlinear RNNs must exhibit an exponentially decaying memory.
> Our result shows that the exponentially decaying memory is an inherent limitation of the architecture.
> The vanishing/exploding gradient phenomenon is not the only issue for learning long-term memory, and improving optimization itself is not sufficient to resolve the memory issue.
> Furthermore, in Section 3.4 first paragraph (Page 7), we identify reparameterization of recurrent weights as a feasible solution to resolve the memory limitation of recurrent architecture.

---

> ### Author Response · Authors · 2023-11-18
>
> 2. `I believe that the authors need to consider giving more insights from their results for a broader context. The stable parameterization section is not very convincing. Can you give a more concrete example about the effects of the reparameterization? Possibly, using a dynamical system or real data?`
>
> **The learning of long-term memory requires the largest eigenvalue real parts of recurrent weights converging to 0 (from negative direction).**
>
> The effects of reparameterization are two-fold.
> - For approximation, based on the proof of Bernstein theorem, we argue that stable parameterization can be used to resolve the memory limitation of nonlinear RNNs.
>     In the first paragraph of Section 3.4 (Page 7), we show that the stable parameterization enables the radius-$\beta$ stability without forcing the eigenvalues of recurrent weights to be bounded away from 0.
>     **Without reparameterization, the stability requires the eigenvalues to be bounded away from 0. With stable parameterization, the stability can be achieved without enforcing the eigenvalues to be bounded away from 0.**
>     A broad class of parameterizations, capable of overcoming the intrinsic memory limitation of the nonlinear RNN architecture, has been identified in Section 3.4.
> - For optimization, reparameterization is used to improve the optimization stability in linear time-invariant system and state-space models.
>     The exponential and softplus parameterizations have been adopted for the stability of linear RNN layer in state-space models [3,4,5].
>
> Any linear time-invariant dynamical system corresponds to a linear functional.
> The synthetic dataset of linear functional with polynomial memory $y_t = H_t(\mathbf{x}) = \int_{-\infty}^t \rho(t-s) x_s ds,$ $\rho(t-s) = \frac{1}{(t+1)^{1.5}}$ serves as a dynamical system example with long-term memory.
> In Figure 2 and Figure 4, we show that RNNs without reparameterization cannot learn target with polynomial memory and RNN using exponential parameterization and softplus parameterization can stably approximate the polynomial memory targets.
> The details of this dynamical system example are given in Appendix D.
> **In Appendix E, we further demonstrate that three stable parameterizations improve the learning of long-term memory**, as evidenced by accelerated optimization in image classification tasks on MNIST.
> Compared to scenarios without reparameterization, these stable parameterizations demonstrate better final validation and test accuracies over three repeats:
>
> |         | Test loss(std err)      | Test accuracy (std err)|
> |---------|-------------------------|--------------|
> | softplus (stable)| 0.8903 (0.0004)         | 71.36(0.03)  |
> | exp (stable)     | 0.9028 (7.2E-05)        | 70.55(0.01)  |
> | inverse (stable) | 0.9082 (0.0005)         | 70.77(0.05)  |
> | direct (unstable)  | 0.9177 (0.0105)         | 68.47(0.06)  |
>
>
> ---
>
> References
>
> [1]. Bengio, Yoshua, Patrice Simard, and Paolo Frasconi. "Learning long-term dependencies with gradient descent is difficult." IEEE transactions on neural networks 5, no. 2 (1994): 157-166.
>
> [2]. (Vanishing gradient) Hochreiter, Sepp. "The vanishing gradient problem during learning recurrent neural nets and problem solutions." International Journal of Uncertainty, Fuzziness and Knowledge-Based Systems 6, no. 02 (1998): 107-116.
>
> [3]. (LRU) Orvieto, Antonio, Samuel L. Smith, Albert Gu, Anushan Fernando, Caglar Gulcehre, Razvan Pascanu, and Soham De. "Resurrecting recurrent neural networks for long sequences." arXiv preprint arXiv:2303.06349 (2023).
>
> [4]. (S4) Albert Gu, Karan Goel, and Christopher Re. "Efficiently Modeling Long Sequences with Structured State Spaces." In International Conference on Learning Representations. 2021
>
> [5]. (S5) Smith, Jimmy TH, Andrew Warrington, and Scott Linderman. "Simplified State Space Layers for Sequence Modeling." In The Eleventh International Conference on Learning Representations. 2022.

---

### Author Response · Authors · 2023-11-18
**Response to all - revision summary**

We thank all the reviewers for their insightful reviews.

The reviewers thought the work **" is the first paper studying the inverse approximation theorem for nonlinear RNNs"**, **"provides a robust proof about exponentially decaying memory"**, **"is well written, in a way that is accessible to a decently large audience"**.
**The notion of stable approximation is an interesting proxy** for how a function can be learned by gradient descent which seems to be rare in approximation theory.

Below we present a summary of the revisions.
In the paper, the revision has been marked in blue.

1. In Appendix E, we add the experiment on MNIST to show that three stable reparameterizations improve the learning of long-term memory. It can be seen in Figure 9 that both the training speed and the test accuracy is improved with the stable parameterization.
2. In Appendix F, we show the feasibility to use impulse inputs to evaluate the memory function and give a proof sketch for the Theorem 3.9.
3. In Appendix G, we show the feasibility to use reversed inputs (suggested by reviewer dqeg) to evaluate the memory function and give a corresponding proof sketch for the Theorem 3.9.
4. In Appendix H, numerical verification is given for the "equivalence" of memory functions evaluated over Heaviside inputs and impulse inputs. The similarity indicates the memory function is a good surrogate tool for the study of memory patterns.
5. In Appendix I.1, as a supplementary content to Figure 3, we present examples of randomly initialized nonlinear RNNs having non-decaying memory.
    In Appendix I.2, we conduct further experiments to numerically demonstrate Theorem 3.9. Transformer does not have a decaying memory thus the stable approximation cannot be achieved by nonlinear RNNs.
6. In Appendix J, we apply the proof technique of Theorem 3.9 to analyze the memory pattern of GRU. It shows the feasibility to apply our methods for more complex models. More techniques are needed to analyze the complex hidden state dynamics, which will be the direction of future research.

---

### Meta-Review · Area_Chair_u7Nt · 2023-12-09

**Metareview:**

The paper presents a theoretical result for nonlinear recurrent neural networks (RNNs), characterizing the types of functions that can be approximated with RNNs. It shows that if a a target sequence can be approximated by a RNN, then it has and exponentially decaying memory. The reviewers agree that the paper is well written and presents an interesting result. The major concern by one of the reviewers was that it might be out of scope of ICLR, because of the lack of practical insights. I disagree with this statement and the authors have provided a good rebuttal to this point. Hence, I recommend acceptance.

**Justification For Why Not Higher Score:**

Excellent contribution but potentially narrow in scope.

**Justification For Why Not Lower Score:**

Strong theoretical developments.

---

### Decision · Program_Chairs · 2024-01-16

Accept (spotlight)